# Structural basis of peptidoglycan synthesis by *E. coli* RodA-PBP2 complex

Rie Nygaard[1], Chris L. B. Graham[2], Meagan Belcher Dufrisne[3], Jonathan D. Colburn[2,4], Joseph Pepe[1], Molly A. Hydorn[5], Silvia Corradi[1,6], Chelsea M. Brown[2,4], Khuram U. Ashraf[1], Owen N. Vickery[2,4], Nicholas S. Briggs[2], John J. Deering[2], Brian Kloss[7], Bruno Botta[6], Oliver B. Clarke[1,8], Linda Columbus[3] ✉, Jonathan Dworkin[5] ✉, Phillip J. Stansfeld[2,4] ✉, David I. Roper[2] ✉ & Filippo Mancia[1] ✉

Peptidoglycan (PG) is an essential structural component of the bacterial cell wall that is synthetized during cell division and elongation. PG forms an extracellular polymer crucial for cellular viability, the synthesis of which is the target of many antibiotics. PG assembly requires a glycosyltransferase (GT) to generate a glycan polymer using a Lipid II substrate, which is then crosslinked to the existing PG via a transpeptidase (TP) reaction. A Shape, Elongation, Division and Sporulation (SEDS) GT enzyme and a Class B Penicillin Binding Protein (PBP) form the core of the multi-protein complex required for PG assembly. Here we used single particle cryo-electron microscopy to determine the structure of a cell elongation-specific *E. coli* RodA-PBP2 complex. We combine this information with biochemical, genetic, spectroscopic, and computational analyses to identify the Lipid II binding sites and propose a mechanism for Lipid II polymerization. Our data suggest a hypothesis for the movement of the glycan strand from the Lipid II polymerization site of RodA towards the TP site of PBP2, functionally linking these two central enzymatic activities required for cell wall peptidoglycan biosynthesis.

Cell shape in bacteria is determined and maintained by the extracellular polymer peptidoglycan (PG), a mesh-like sacculus surrounding the cytoplasmic membrane composed of polymerized glycan chains cross-linked by short peptides[1]. PG synthesis is rate limiting for bacterial growth, and its disruption results in cell lysis or cessation of growth as exploited by many natural product and semisynthetic antibiotics[2–5]. These include β-lactams, the most clinically successful antibiotics to date[6,7]. The cytoplasmic proteins that synthesize the PG precursor, Lipid II – an undecaprenyl ($C_{55}$) pyrophosphate (Und-PP)-linked disaccharide of *N*-acetylglucosamine (GlcNAc) and *N*-acetylmuramic acid (MurNAc)-pentapeptide – and the extracellular proteins responsible for the subsequent polymerization of PG, have been individually characterized, biochemically and structurally[8,9].

In the periplasm, PG biosynthesis begins with a Lipid II-specific glycosyltransferase (GT) which forms a glycan strand polymer by linking the disaccharides of two Lipid II molecules, one termed the

[1]Department of Physiology and Cellular Biophysics, Columbia University Irving Medical Center, New York, NY 10032, USA. [2]School of Life Sciences, University of Warwick, Coventry CV4 7AL, UK. [3]Department of Chemistry and Department of Molecular Physiology and Biological Physics, University of Virginia, Charlottesville, VA 22904, USA. [4]Department of Chemistry, University of Warwick, Coventry CV4 7AL, UK. [5]Department of Microbiology and Immunology, Columbia University Irving Medical Center, New York, NY 10032, USA. [6]Faculty of Pharmacy and Medicine, Sapienza University of Rome, Rome, Italy. [7]New York Consortium on Membrane Protein Structure, New York Structural Biology Center, 89 Convent Avenue, New York, NY 10027, USA. [8]Department of Anesthesiology, Columbia University Irving Medical Center, New York, NY 10032, USA. ✉e-mail: columbus@virginia.edu; jonathan.dworkin@columbia.edu; phillip.stansfeld@warwick.ac.uk; David.roper@warwick.ac.uk; fm123@columbia.edu

donor and the other the acceptor, and thereby releasing Und-PP from the donor site (Fig. 1a). After the two initial Lipid II molecules have been linked together, the resulting tetra-disaccharide attached to Und-PP, termed Lipid IV, becomes the donor for another Lipid II acceptor, in turn linking its tetra-saccharide to the Lipid II di-saccharide to yield

Lipid VI. This cycle repeats in a processive manner creating progressively longer polysaccharide chains attached to Und-PP (the roman numeral denotes the number of monosaccharide groups in the polysaccharide chain). Once the growing glycan polymer reaches a sufficient length, it is attached to the existing PG sacculus via peptide

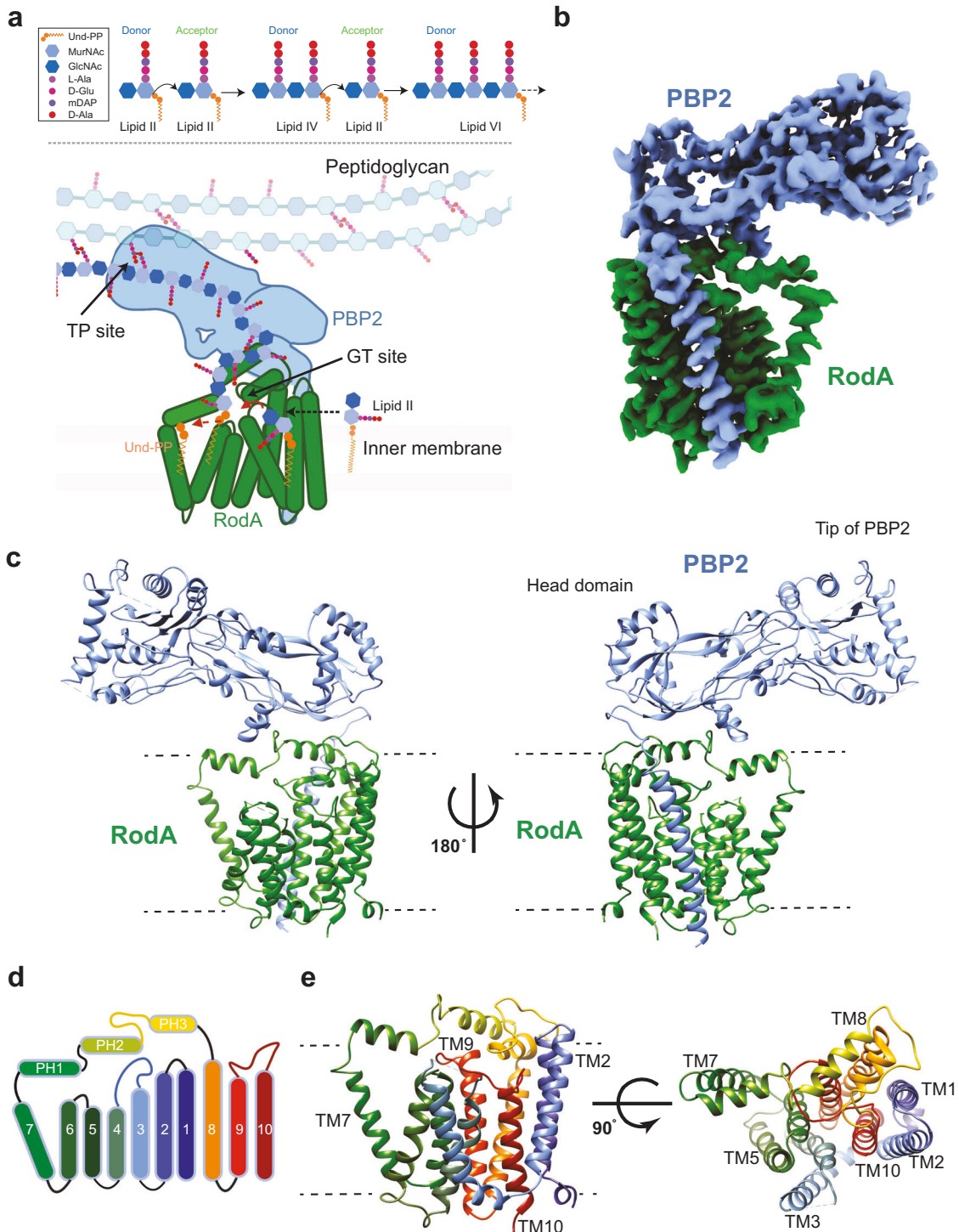

**Fig. 1 | Mechanism and overall structure of RodA-PBP2 complex. a** Above, schematic representation of the reaction catalyzed by RodA (green) and PBP2 (blue). Below, the chemical representation of peptidoglycan synthesis with Lipid II building blocks and chemical components of Lipid II are shown in a box on the top left. **b** Cryo-EM density map of the RodA-PBP2 complex. Density corresponding to RodA and PBP2 is shown in green and blue, respectively. **c** Structure of the RodA-PBP2 complex shown as a ribbon with RodA in green and PBP2 in blue. Approximate membrane boundaries are represented as dotted lines. **d** Schematic diagram showing the topology of RodA colored in rainbow from C (blue) to N terminus (red), consisting of ten TM helices and a well-ordered periplasmic region between TM helix 7 and 8. **e** Structure of RodA shown rotated 90°, with the ten TM helices colored as in **e**. Approximate membrane boundaries are again shown as dotted lines.

crosslinks between the pentapeptide of the glycan strand and a peptide stem on the existing PG sacculus by a transpeptidase (TP) to yield crosslinked PG (Fig. 1a). In *E. coli*, the GT RodA from the Shape, Elongation, Division, and Sporulation (SEDS) family, and PBP2, the monofunctional TP class B Penicillin Binding Protein, mediate these respective enzymatic tasks[10]. RodA is an integral membrane protein consisting of ten transmembrane (TM) helices[11], while PBP2 has a single TM helix and an extracellular domain with a classical class B PBP fold containing the TP active site[12,13]. Together, they comprise the core of the elongasome[14], the complex responsible for the determination of bacterial rod shape. Despite recent advances in our understanding of this molecular machine[15,16], not least as derived from the crystal structure of a *Thermus thermophilus* RodA-PBP2 complex[17], fundamental mechanistic questions remain unresolved. These include the characterization of molecular determinants and conformational states required for i) Lipid II binding, ii) GT polymerization of glycan strands, and iii) subsequent translocation of the glycan polymer to the TP active site.

Here, we used single-particle cryo-electron microscopy (cryo-EM) to determine the structure of the *E. coli* RodA-PBP2 complex, expressed as a functional fusion and reconstituted in lipid-filled nanodiscs, to 3.0 Å resolution. We used an integrated approach – combining structural information with biochemical and genetic assays, molecular dynamics (MD) simulations and site-directed spin labeling (SDSL) double electron-electron resonance (DEER) experiments – to investigate the molecular determinants of substrate binding and catalysis, and conformational changes required for this processive machinery to function. Our studies suggest a mechanism that would facilitate the migration of a growing glycan polymer towards the TP site of PBP2, enabling TP-dependent crosslinking to form the PG layer.

## Results

### Structure determination of RodA-PBP2

Although most bacteria contain separate open reading frames (ORFs) encoding the GT (SEDS) and TP (PBP) activities, examples exist of a single ORF encoding a SEDS-PBP fusion[18]. We previously demonstrated that a synthetic construct consisting of a SEDS protein fused to its cognate PBP functionally complements gene deletions of both enzymes[13]. We reasoned that a SEDS-PBP fusion would be biochemically more tractable than the individual components, and that its structure in a lipid environment would greatly facilitate a mechanistic understanding of the overall complex. We screened 189 SEDS orthologs for expression and stability in detergents to identify those amenable for structural studies[19]. From this initial screen, SEDS proteins from five different species with the highest expression levels and stability were selected to design a set of fusions with one or more species-matched PBPs, resulting in eight unique constructs (Supplementary Table 1). These were again evaluated for expression and stability in detergent after metal-affinity chromatography (using a genetically encoded poly-histidine tag), and mono-dispersity as assessed by size-exclusion chromatography (SEC). Based on these criteria, a fusion construct between *E. coli* RodA and PBP2 (Supplementary Fig. 1a) had the most promising profile for structure determination.

Next, we confirmed that this *E. coli* RodA-PBP2 fusion has GT activity – which is not inhibited by moenomycin, in contrast to class A bifunctional PBPs – as observed by polymerization of Lipid II modified with a fluorescent dansyl group (dansyl lysine Lipid II; Supplementary Fig. 1b, c, e). The TP domain binds (and is covalently modified by) a fluorescent penicillin mimic bocillin[20] Supplementary Fig. 1d, f), suggesting that the TP domain is intact. We purified the RodA-PBP2 fusion protein, and as controls RodA alone (terminating at residue 373 of RodA) and a fusion of RodA with only the single TM helix of PBP2 (terminating at residue 47 of PBP2) in detergent (Supplementary

Fig. 1g). These proteins were assayed for their Lipid II polymerization activity (Supplementary Fig. 1h). RodA in isolation has residual GT activity but is significantly stimulated by the presence of the transmembrane helix of PBP2 in both the full length fusion and the truncated version. These results are consistent with what was previously shown for the *T. thermophilus* proteins[17].

The RodA-PBP2 fusion was purified to homogeneity in detergent by metal-affinity chromatography followed by SEC, and then reconstituted into lipid-filled nanodiscs (Supplementary Fig. 1i-l) for subsequent vitrification and cryo-EM analysis. This resulted in two maps locally refined around the TM and periplasmic regions, both at 3.0 Å resolution (Fig. 1b, Supplementary Table 2 and Supplementary Fig. 2). In the TM map, we could reliably build the RodA structure, assigning the sequence from residues 9 to 93 and 109 to 364, as well as the adjacent PBP2 single TM helix from residue 10 to 40 (Supplementary Fig. 3). The structure shows ten RodA TM helices, with intracellular N- and C-termini. TM helices 1-6 and TM helices 8–10 form a tight helical bundle, with TM helix 7 extending away from it, stabilized by three periplasmic juxtamembrane helices (PH1, PH2 and PH3) (Fig. 1d, e). Additionally, the PBP2 single TM helix, is ordered and packs against RodA TM helices 8 and 9 (Fig. 1c).

The PBP2 soluble domain is less well resolved, but using Namdinator[21] and its previously published crystal structure as the input model[22], we were able to build residues 10–343, 401–430, 457–540 and 569–612. We were unable to observe interpretable density for residues 344–400 and 431–456 at the tip of PBP2, proximal to the TP active site (Supplementary Fig. 4a). The binding site for MreC[23] – a scaffolding protein that binds to PBP2 and is thought to impact its activity[23,24] – also known as the head domain, has the lowest local resolution in the map (Supplementary Fig. 4b). Nevertheless, the resolution was sufficient to fit the helices in the head domain to the density map, despite the fact that we observe substantial structural changes compared to the previously published X-ray crystal structure of PBP2 from *E. coli*[22] (Supplementary Fig. 4a).

Comparing the *T. thermophilus* model[17] with our model, we observe that the overall structure of the TM region is very similar between the two, despite having a sequence identity of only 39% (Supplementary Fig. 4c). This includes the positioning of TM helix 7, which in both extends similarly away from the helical TM core. In contrast, in the periplasmic domain of PBP, we observe quite substantial structural differences. There is both a relative turn and a tilt between the two, and a closing of the head and anchor domain in our structure as opposed to an opening in the *T. thermophilus* model (Supplementary Fig. 4c).

### Putative substrate binding cavities

Synthesis of the glycan polymer requires an acceptor and a donor binding site in RodA, both initially accommodating Lipid II. The two initial Lipid II substrates are linked via a GT reaction, coupling the MurNAc sugar of the Lipid II in the donor site to the GlcNAc of the Lipid II in the acceptor site, with Und-PP as the product in the donor site (Fig. 1a). Since RodA is a processive enzyme, Lipid IV (polymerized Lipid II after the first reaction) now becomes the donor so that the MurNAc directly bound to Und-PP in Lipid IV can be transferred to an incoming Lipid II acceptor. For this to occur, Lipid IV needs to transition from the acceptor to the donor sites. This movement of the growing glycan chain from the acceptor to the donor site is processive, thus allowing the cycle to repeat producing progressively longer chains (i.e., Lipid VI, Lipid VIII, etc.).

Analysis of the RodA portion of the structure for putative substrate-binding sites reveals two major cavities (termed cavity A and B) (Fig. 2a). Cavity A is located between TM helices 6, 7 and 9, and is framed on one side by PH1 and on the other by TM helices 5 and 6 and the periplasmic loop (PL3) connecting the two. The rim of cavity A is lined with conserved residues (Fig. 2a) and overall is polar in nature

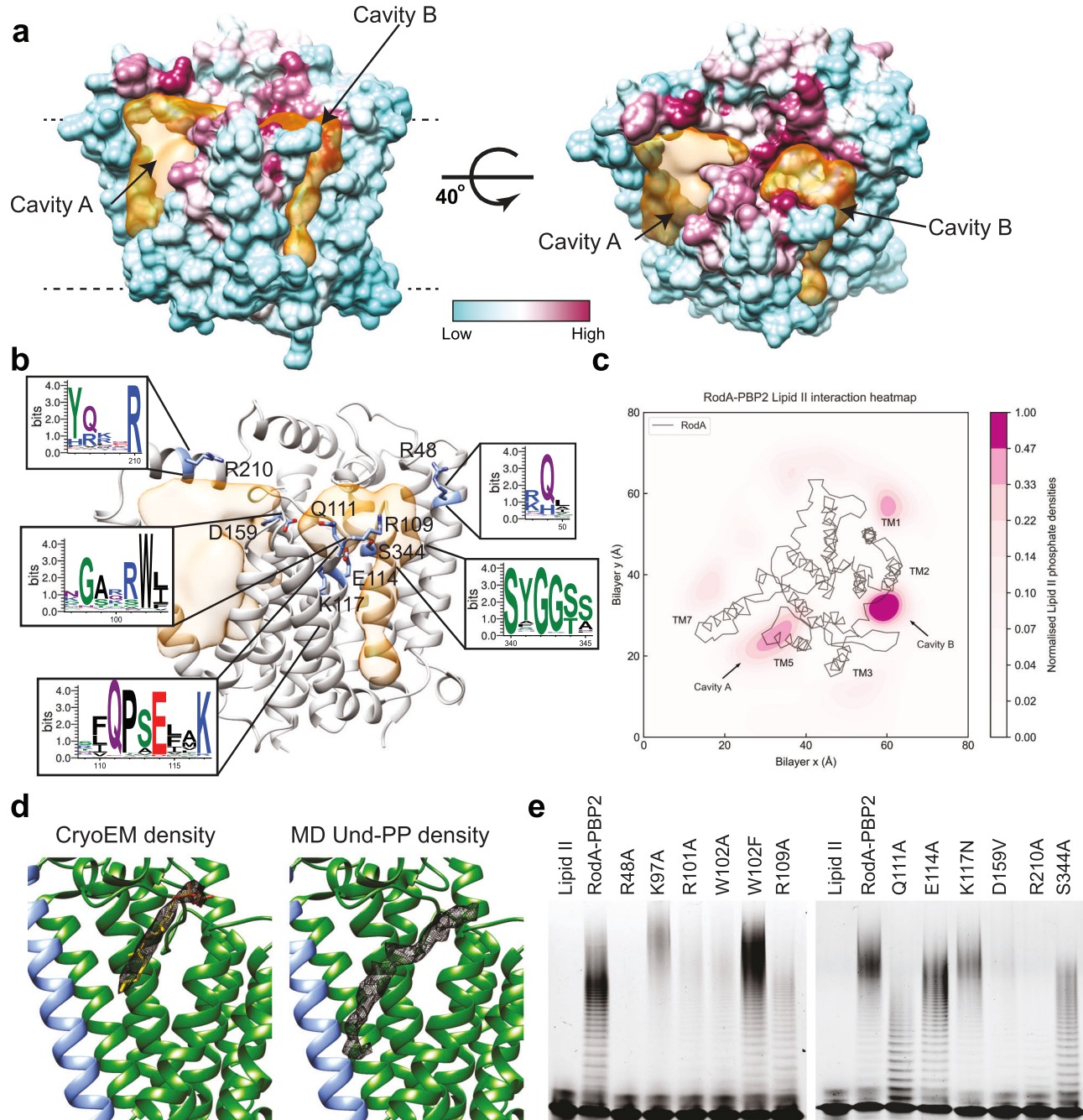

**Fig. 2 | Lipid II binding cavities. a** Cavity analysis of the RodA-PBP2 structure with RodA-PBP2 shown as surface colored by conservation on a green (no conservation) to purple (absolute conservation) scale. The cavities are shown as a semi-transparent surface in orange. Volumes were calculated using the Voss Volume Voxelator (3 V) server[52] using probes with 10 and 2 Å radii, corresponding to the outer and inner probe, respectively. **b** Structure of the transmembrane region of RodA-PBP2 shown as a ribbon with residues of interest shown as sticks, and the cavities shown as in **a**. Cut-outs from Weblogo plots[58], as in Supplementary Fig. 6, shown for regions of interest. **c** Density plot of Lipid II from 50 repeats of 10 μs in unbiased CG MD simulations of the RodA-PBP2 complex. The plot shows two preferred binding sites of Lipid II overlapping with cavity A and cavity B. **d** Comparison of (left) lipid-like density observed in cavity A in the cryo-EM map (only the region of the map corresponding to the lipid like density is shown) with (right) average density of Und-PP from unbiased atomistic MD simulation of RodA-PBP2 with Und-PP initially docked into cavity A. **e** Functional analysis of RodA-PBP2 GT activity using in vitro Lipid II polymerization assay where the effect of different residues was studied by introducing point mutations and testing for activity. Source data are provided as a Source Data file.

(Supplementary Fig. 4d). It extends into the membrane, where it becomes exposed to the lipid bilayer, and is lined with hydrophobic residues (Fig. 2a and Supplementary Fig. 4d). Cavity B is located on the opposite side of RodA relative to cavity A, between TM helices 2, 3, 4 and 10 and is also exposed to the membrane (Fig. 2a). Residues lining cavity B are less conserved, but as for cavity A we observe several

positively charged residues on its periplasmic side (Fig. 2a and Supplementary Fig. 4d). The region connecting the two cavities has some of the highest degree of conservation in the entire structure (Fig. 2a, b). Given their orientation in respect to the TP site of PBP2, we suggest that cavities A and B are the donor and acceptor sites for the two substrates, respectively[24].

To probe this hypothesis, we performed 50 repeats of 10 µs unbiased coarse-grained (CG) MD simulations to identify the location and interactions the two Lipid II substrates have with the surface of RodA. Analysis of these simulations identified two preferred binding sites for Lipid II that directly map to cavities A and B (Fig. 2c). Of the two binding sites, cavity B appears to have higher particle density for Lipid II than cavity A over the total simulation time. This is consistent with cavity B being the acceptor site, continuously recruiting Lipid II from the membrane. The density analysis also revealed further sites of interaction, but when evaluated with PyLipID[25], cavity B, followed by cavity A, had the highest occupancies by Lipid II during the simulations (Supplementary Fig. 5a-d). Specifically, over the CG simulations, Arg48 and Arg109 (cavity B) and Arg210 (cavity A) displayed the highest occupancy of interactions with Lipid II (Supplementary Fig. 5b). In all cases, the predominant protein-Lipid II interactions were with the peptidoglycan-pyrophosphate headgroup, while the lipid tail engaged with both protein and surrounding lipid membrane. We combined these data with ligand docking to refine our model for how Lipid II binds to both cavities A and B (Supplementary Fig. 5e-f).

In agreement with our simulations, we observed an elongated density in the cryo-EM map within cavity A (Fig. 2d). The shape of the density is consistent with Und-PP, an obligate product of each GT reaction (Fig. 1a). To further elaborate on this observation, we ran three repeats of 1 µs atomistic MD simulations of RodA-PBP2 inserted into a phospholipid membrane with Und-PP docked into cavity A (Supplementary Fig. 5g). During these experiments, we observed that Und-PP stayed tightly associated within this cavity and that its average occupancy overlapped with the density present in the cryo-EM data (Fig. 2d).

## Biochemical characterization of residues in and around cavity A and B

Both cavities have positively charged residues on their periplasmic side (Arg210 in cavity A; Arg48 and Arg109 in cavity B) (Fig. 2b and Supplementary Fig. 6). Our simulations suggest that these residues interact with the pyrophosphate group of Lipid II. An equivalent of Arg210 on PH1 appears to be part of a common structural element for GT-C glycosyltransferases that use Und-PP as a carrier, as for example the O-antigen ligase WaaL[26,27]. To test whether Arg210 is essential for function, we mutated it to alanine and found that GT activity in vitro was significantly reduced (Fig. 2e and Supplementary Fig. 7a). We also examined the phenotype of a mutation in the corresponding arginine in *B. subtilis* using an in vivo sporulation assay[13]. Here, spore heat resistance is dependent on the function of the RodA homolog SpoVE (Supplementary Fig. 8) that synthesizes spore PG. We observed that mutating SpoVE Arg212 (corresponding to *E. coli* RodA Arg210) to alanine had a severe effect on sporulation (Supplementary Table 3). This result is consistent with the requirement of this arginine for GT activity (Fig. 2e). Arg48 and Arg109 both point towards cavity B (Fig. 2b). GT activity is severely reduced in vitro and in vivo when Arg48 is mutated to alanine (Fig. 2e, Supplementary Fig. 7a and Table 3) whereas it is only moderately affected in vitro for an Arg109Ala mutant. These results are consistent with an important role of Arg48 and to a lesser extent Arg109, in coordinating the pyrophosphate of Lipid II within or entering cavity B.

The twenty amino acid periplasmic loop connecting TM helices 3 and 4 (PL2) is positioned adjacent to cavity B and could potentially reach from one cavity to the other and interact with the substrates in both cavities, and/or play a role in their transition from the acceptor to the donor sites. This loop is intrinsically flexible, as observed in our MD simulations (Supplementary Fig. 5b), and we could only partially assign the sequence to the density in this region of the cryo-EM map. PL2 has several highly conserved residues, including Trp102 and Gln111, that are invariant in all species analyzed (Supplementary Fig. 6). PL2 also contains two positively charged residues, Lys97 and Arg101, which

while only moderately conserved, are appropriately positioned to engage with Lipid II. To probe the function of PL2, we mutated these two charged residues to alanine. Only Arg101, which is the more conserved of the two, showed significant reduction in activity compared to wild type (Fig. 2e and Supplementary Fig. 7a). Next, we mutated Trp102 or Gln111 to alanine and showed that both mutants had reduced GT activity (Fig. 2e and Supplementary Fig. 7a). However, the GT activity of Trp102 appeared to be maintained with a mutation to phenylalanine. Mutation of the equivalent Trp104 in *B. subtilis* SpoVE to alanine or phenylalanine severely affected the sporulation phenotype (Supplementary Table 3), consistent with the functional importance of this residue.

Finally, we probed the potential roles of other highly conserved residues located in or between the two cavities (Glu114, Lys117, Asp159 and Ser344) (Fig. 2b). GT activity in our in vitro GT assay was not affected when Ser344 was mutated to alanine whereas we observed a reduction in activity when Asp159 was mutated to valine, consistent with the severe effect of a mutation of *B. subtilis* SpoVE Asp163 (the equivalent of *E. coli* Asp159) to valine in our in vivo sporulation assay (Supplementary Table 3). Finally, while mutations of Glu114 to alanine and Lys117 to asparagine had only a modest impact on GT activity in vitro (Fig. 2e and Supplementary Fig. 7a), identical mutations of the corresponding residues in *B. subtilis* SpoVE (Glu116Ala, Lys119Asn) severely affected sporulation efficiency (Supplementary Table 3). The differences observed between the in vivo and in vitro assays suggest that even modestly reduced in vitro activity may not be sufficient for maintaining in vivo function, and highlight the importance of characterizing mutant phenotypes in vivo.

## Conformational flexibility within RodA

In order for the glycan strand to form and extend towards the TP active site of PBP2, the growing Und-PP-linked glycans (Lipid II, IV, VI, etc.) must transition from the acceptor site (cavity B) to the donor site (cavity A) site at each round of catalysis[28]. For this to occur, a conformational change in RodA must take place to open a passageway between TM helices 1-2 and 8-10 on one side, and the helical bundle of TM helices 3-7 on the other (Fig. 3a). The cryo-EM density map reveals that the bundle composed of TM helices 3-7 has an overall lower resolution than the rest of the protein, suggesting some degree of flexibility within this region (Fig. 3b).

We used SDSL DEER spectroscopy to further probe this hypothesis. The resulting DEER-derived probability distribution of distances between two nitroxide spin labels (MTSL; R1) introduced at specific sites provides information about the number and population of conformational states, as well as a distance restraint for each one. We replaced the two native cysteines in RodA by mutagenesis to glycine and alanine (Cys82Gly and Cys133Ala) and confirmed that the cysteine-free fusion RodA-PBP2 complex was expressed and functional (Supplementary Fig. 9a, b). We then introduced a spin label pair – Gly44R1 on the periplasmic end of TM helix 2 and Asp90R1 on the periplasmic end of TM helix 3 – in the RodA fusion with the cysteine-free background (Fig. 3c and Supplementary Fig. 9a-c). From the resulting DEER-derived distance distribution, three populations were observed with dominant distances at 26 Å, 35 Å, and 45 Å (Fig. 3d and Supplementary Fig. 9d, e). Predicted distances calculated by adding sterically allowed MTSL rotamers to the cryo-EM structure in silico yielded a distance distribution between 34 and 48 Å (Fig. 3d), aligning best with the longest experimental distance population. Side chain rotamers, which can contribute up to +/- 8 Å (and accounted for in the in silico modeling) may be able to partially explain the middle-distance population (centered at 35 Å). However, a backbone conformational change from the cryo-EM structure is needed to sample the majority of the distances between 20 and 35 Å, implying that multiple conformations of the helical bundle exist. These populations have shorter distances between Gly44R1 and Asp90R1 than observed in the cryo-EM map so

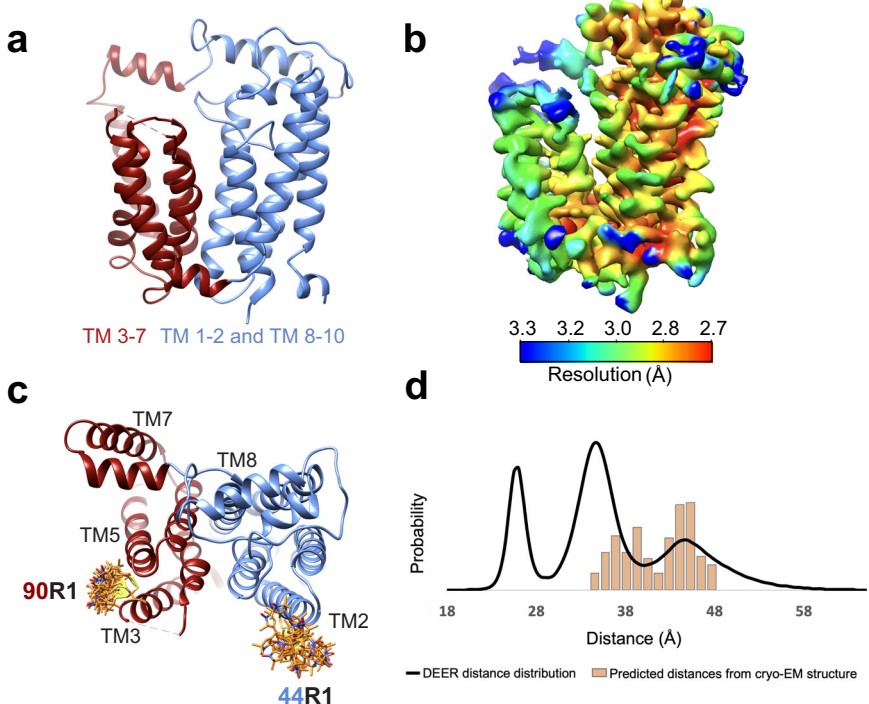

**Fig. 3 | Dynamics within the RodA. a** RodA shown as ribbon colored by domain with TM helices 3-7 red and TM helices 1-2 and TM helices 8-10 blue. **b** Local resolution of the transmembrane region of the RodA-PBP2 complex from the final local resolution refinement with a mask around the transmembrane part of the complex. It is apparent that the domain consisting of TM helices 3-7 overall has lower local resolution. **c** Sterically allowed rotamers of nitroxide spin labels (R1) are illustrated for Gly44R1/Asp90R1 on the cryo-EM structure (orange sidechains). **d** Resulting DEER distance distributions (solid black line) are compared to predicted distance distributions (bars) based on the sterically allowed rotamers shown in the structures (colored as sidechains).

further investigation is needed to determine the structure and physiological relevance of these states.

### The RodA active site and mechanism of catalysis

Within the highly conserved region between the two cavities and centrally located in periplasmic loop 4 (PL4) is Asp262, previously identified as a catalytic residue[15,17,29] (Fig. 4a). To further define the active site, we performed systematic mutagenesis of residues surrounding Asp262, and tested their effect on function via our biochemical assay. Consistent with a previous report[10], mutation of Asp262 to alanine renders RodA-PBP2 enzymatically inactive (Fig. 4b and Supplementary Fig. 7a). Mutations Glu258Ala, His260Ala or Thr261Ser did not affect activity despite the high degree of conservation and spatial-proximity of these residues to Asp262, whilst mutation of Pro257 – also proximal to Asp262 – to alanine resulted in complete loss of GT activity (Fig. 4b, Supplementary Fig. 6 and 7a). A similar severe phenotype is observed when the equivalent of Asp262 is mutated in our in vivo *B. subtilis* assay (Asp263Ala) together with a more modest reduction in activity for the equivalent of Glu258 (Glu259Ala) (Supplementary Table 3).

Based on our modelled and simulated coordinates of Lipid II bound to both cavities (Supplementary Fig. 5a), we propose a mechanism, in which Asp262 plays a central role in enabling the GT reaction to occur (Fig. 4c), analogous to His338 in WaaL[26]. In our scheme, Asp262 abstracts a proton from the 4' hydroxyl of Lipid II in cavity B. This abstraction of the proton would then allow the 4' oxygen to perform a nucleophilic attack on the 1' carbon of MurNAc of Lipid II in cavity A. This breaks the bond between the carbon and the oxygen of the phosphate resulting in a Und-PP product in cavity A and Lipid IV (or Lipid VI, Lipid VII etc.) in cavity B. Our MD simulations suggest that Arg210 in cavity A and Arg48 in cavity B, as well as the arginine residues in PL2 (97-111) (Arg101 and Arg109) engage with the pyrophosphate

head groups of Lipid II to coordinate of the substrate within the active site. We tested whether the reaction is metal dependent, but observed no change in activity by adding the divalent-ion chelator EDTA during the assay, indicating that the GT reaction catalyzed by the *E. coli* RodA-PBP2 fusion is metal-independent (Supplementary Fig. 7b). This contrasts with the GT51 family of enzymes, which includes class A PBPs[30–32].

We probed the feasibility of the proposed mechanism by performing semi-empirical Density Functional Tight Binding (DFTB)[33] calculations on a cluster model of the putative RodA active site (Supplementary Fig. 5a). From these calculations, we observed that Asp262-activated formation of Lipid IV from two Lipid II substrates bound in cavities A and B is plausible given the geometric constraints of the RodA structure as sampled by our atomistic simulations. By using partially-converged nudged elastic band calculations, we present the reaction visually within Supplementary Movie 1. While this does not validate the energetic feasibility of the proposed mechanism, it does illustrate that the coordination of the bound substrates within the active site is geometrically suitable to enable the reaction.

### Formation of a glycan chain and movement of polymer to TP site

The processive mechanism of RodA involves a the shuttling of substrates from the acceptor to the donor site to progressively extend the growing glycan strand, until the leading pentapeptide stem reaches the TP site of PBP2[34,35]. Our structure contains a groove that extends from the extracellular surface of the RodA GT site to the PBP2 TP active site (see below). Ligand docking and modelling different lengths of the glycan strand reveal that Lipid XX (i.e., ten disaccharides) bound to cavity A is of sufficient length to reach the TP active site and for the peptide stem to connect to the already existing PG. We have modelled RodA-PBP2 bound sequentially to Lipid II, Lipid IV, up to Lipid XXII and MD simulations show that the polysaccharides are stably coordinated

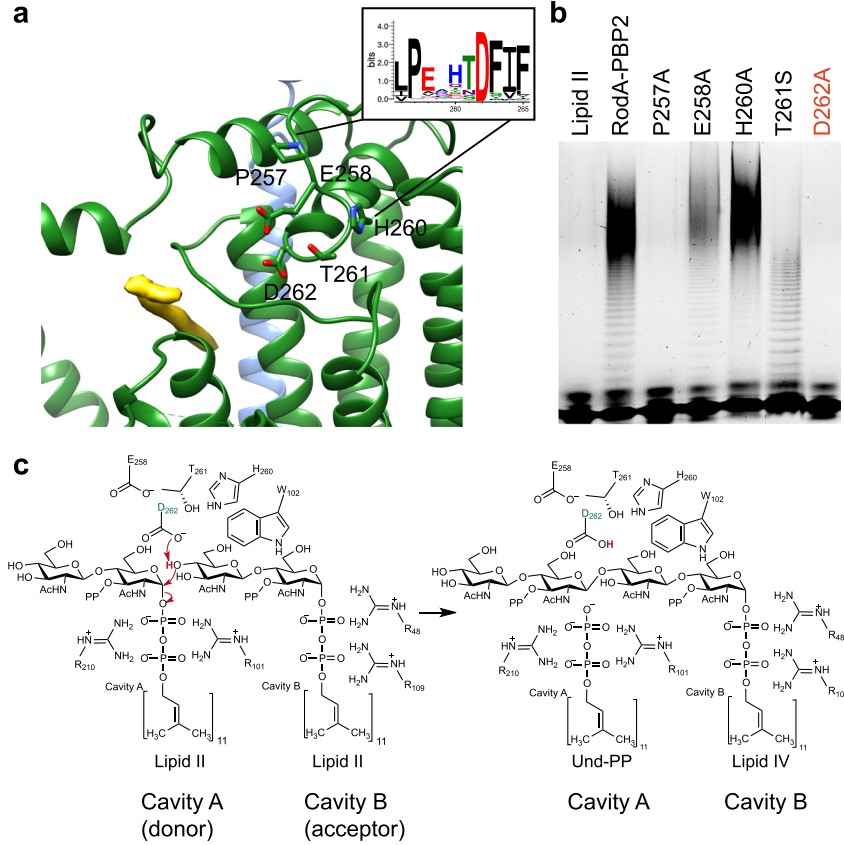

**Fig. 4 | Mechanism of Lipid II transglycosylation. a** RodA active site with RodA shown as green ribbon and PBP2 shown as blue ribbon, residues of interest represented as sticks. Cut-out from Weblogo plot, as in Supplementary Fig. 6, shown for residues of interest, residues are represented as sticks. Cryo-EM density in cavity A proposed to correspond to Und-PP displayed as a yellow surface. **b** Functional analysis of RodA-PBP2 GT activity using in vitro Lipid II polymerization assay, monitoring the effect of different mutations on selected residues. Source data are provided as a Source Data file. **c** Schematic representation of the mechanistic model for the GT reaction between two Lipid II molecules by RodA viewed from the periplasmic side of the membrane. The active site Asp262 is highlighted in green.

within this groove (Supplementary Fig. 10a). The pentapeptide stem appears more mobile when not bound to the TP site, and the positions of the polyprenyl tails in and around both cavities A and B anchors the nascent PG to the membrane through interactions both within the protein and with the phospholipid membrane (Supplementary Fig. 10a). The periplasmic portion of the complex has a much higher RMSF than the TM domain, however, the secondary structure of RodA-PBP2 is stable during the simulations (Supplementary Fig. 5b and 10b).

### Dynamics between RodA and PBP2 in the complex

While analyzing the cryo-EM data, we observed considerable structural heterogeneity between RodA and PBP2 (Fig. 5a and Supplementary Fig. 9n). To further investigate this, we performed a 3D variability analysis with a mask around PBP2, which has the lowest local resolution and seems to be the most mobile of the two proteins (Fig. 5a and Supplementary Fig. 9n). After separating the particles into 6 clusters, we observe a vertical tilt of ~10° of PBP2 with respect to the bilayer, as well as a rotation around PBP2 (Fig. 5a and Supplementary Fig. 9n). To complement this observation, we ran MD simulations of RodA-PBP2. Consistent with the cryo-EM data, the tip of PBP2, harboring the TP active site, appears to be highly dynamic in these simulations (Fig. 5c). For the simulations with glycan chains included, especially true for those with longer ones, we observe a large conformational change in the extracellular part of PBP2 that swings upwards with the tip of PBP2 pointing away from the membrane (Supplementary Fig. 10c). We do not observe an extended conformation like this in our cryo-EM data, but as discussed above, we observe heterogeneity in the sample and

discarded numerous particles that may adopt more transient conformations of PBP2.

To further study this, we investigated the dynamics between PBP2 and RodA with DEER using an engineered cysteine pair between residue Gly44 at the periplasmic side of TM helix 2 in RodA and residue Gln99 in the head domain of PBP2 (residue 476 in the fusion) (Fig. 5b). PBP2 Gln99 is located in the proposed MreC binding site[15,23,24], the region where we observe the most variability in the cryo-EM data (Supplementary Fig. 9n). This pair (Gly44Cys, Gln99Cys) was labelled with cysteine-reactive nitroxide spin-label (R1; MTSL), confirmed to be functional (Supplementary Fig. 9n), and DEER data were collected. The resulting DEER data for Gly44R1 and Gln99R1 show a broad distribution of distances ranging from 30 to 65 Å (Fig. 5d and Supplementary Fig. 9f-h). Residue Gly44 in RodA is in a relatively static region of the structure (Supplementary Fig. 5b), so most motions contributing to the DEER distances likely arise from the dynamics of the MreC-binding head domain of PBP2 and/or potentially from the dynamic nature of the entire soluble domain of PBP2 with respect to RodA. Sterically allowed MTSL rotamers attached to residues of the cryo-EM structure in silico (Fig. 5b) result in a calculated distance distribution ranging from 18 to 40 Å but do not sample the longer distances of the broad DEER distribution (Fig. 5d). Furthermore, when MTSL rotamers were computationally attached to a structure from MD (50 ns frame from Supplementary Fig. 10c) that has an extended conformation of the head domain of PBP2, the resulting distance distribution was between 46 and 70 Å (Fig. 5d). In comparison, a DEER pair within the PBP2 domain (Lys185R1 and Ser330R1) yields a single more narrow DEER

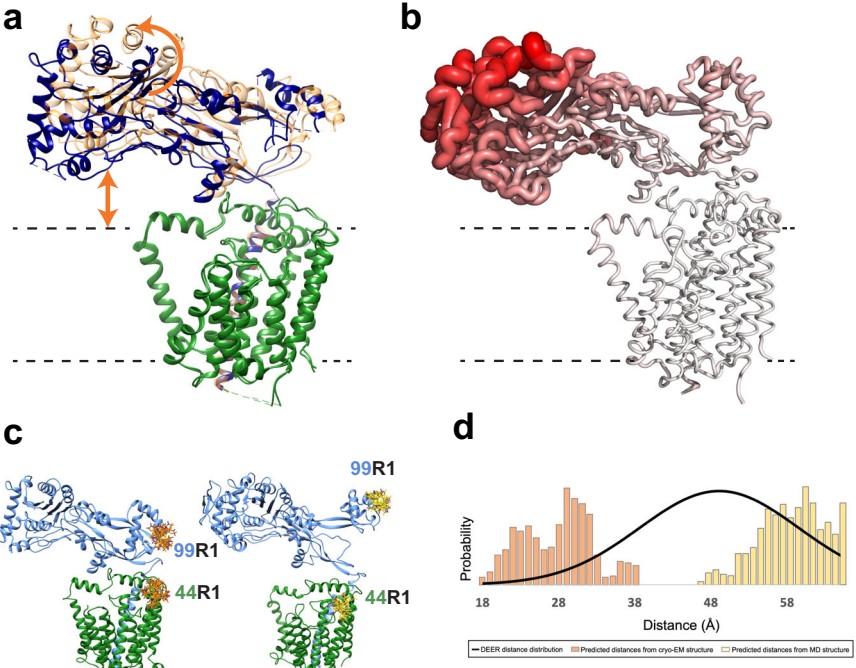

**Fig. 5 | Dynamics within the RodA-PBP2 complex. a** The two extreme conformations of PBP2 from the cryo-EM 3D variability analysis. **b** RodA-PBP2 shown as ribbon colored by RMSF over 3 MD simulations of 1 μs aligned to the RodA backbone, with the ribbon increasing in thickness with greater RMSF. **c** Sterically allowed rotamers of nitroxide spin labels (R1) are illustrated for Gly44R1/Gln99R1 on the cryo-EM structure (left; orange sidechains) and a structure from an MD simulation (right; yellow sidechains). **d** Resulting DEER distance distributions (solid black line) are compared to predicted distance distributions (bars) based on the sterically allowed rotamers shown in the structures (colored as sidechains).

distance distribution (Supplemental Fig. 9i-m). These data support the notion that the flexibility observed between the RodA and PBP2 domains is likely due to movement of the head domain with respect to RodA. Collectively, these data suggest that the DEER distance distribution cannot be explained solely by side-chain conformers or small backbone differences from the cryo-EM structure and must represent substantial backbone or domain conformational changes as observed in the MD simulations.

## Discussion

The GT RodA and the TP PBP2 constitute the core of the *E. coli* elongasome, the multiprotein complex that catalyzes formation of the essential PG layer. In combination, our results provide data-supported hypotheses for the architecture, catalysis, and structural rearrangements required for peptidoglycan synthesis. The cryo-EM structure identified two major cavities (A and B) on the periplasmic side that coarse grained MD analysis supports as two Lipid II binding sites, with cavity A and B as the donor and acceptor site, respectively. In our cryo-EM map, within the putative donor site there is a lipid-like density, which we tentatively assigned to Und-PP, a product of the GT reaction. MD simulations show that Und-PP can stably occupy both cavities and that its average occupancy overlaps well with the observed experimental density.

By combining in vitro and in vivo functional assays that measure GT activity on purified protein and sporulation efficiency in *B. subtilis*, respectively, we have begun to characterize the role of some of the most conserved and/or charged residues in cavity A and B. The structural homology between RodA and other GT-C type glycosyltransferases that utilize Und-PP-coupled ligands[26], suggests that the positively-charged residues could participate in coordinating the pyrophosphate to either facilitate its recruitment into the binding sites or position the substrate for catalysis to occur. Some, such as Arg210, seem to be essential for function and others less so, reflecting perhaps the fact that substrate affinity and specificity are determined by multiple sites of interaction with the protein. Of interest is the conservation and functional significance of a tryptophan at position 102 in PL2 in RodA, which may play a similar role to Trp383 in the bacterial cellulose synthase BscA[36]. In BscA, this tryptophan residue was proposed to act as part of a "finger helix" interacting with the cellulose substrate sugars, enabling a path for the growing glycan strand polymer out of the active site after its formation, which is characteristic of a processive GT enzyme[37,38]. Trp102 in *E. coli* RodA could perform an analogous role in the processive Lipid II polymerization mechanism. In between the two cavities and bridging them, we also find the active site residue Asp262 and several conserved residues delineating the active site. We probed the functional relevance of these residues both in vitro and in vivo, and also showed that the GT reaction is metal-independent. By combining these results with semi-empirical DFT calculations on the putative RodA active site, we propose a mechanism for how RodA catalyzes the reaction between the two Und-PP-linked glycans (Fig. 6).

To enable processivity, the growing Und-PP-linked glycan chain must transition at each catalytic cycle from the acceptor to the donor cavity. Based on the relative flexibility of the structure as observed in the cryo-EM maps, we hypothesize a movement to create a passageway between TM helices 3-7 on one side and 1-2, 8-10 on the other to form a conduit for the lipid tail of Und-PP. We investigated this hypothesis by DEER, inserting probes on cysteine mutants, designed based on the structure and introduced on a cysteine-less background. These experiments yielded results consistent with flexibility of the TM 3-7 subdomain.

Our structure suggests how the growing glycan polymer can extend from the RodA GT site to the TP active site within PBP2 (Fig. 6). By combining modelling of Lipid II, Lipid IV and so forth up to Lipid XX with MD simulations, we determined that that the growing glycan is stable within the groove leading to the TP site, that the polyprenyl tail can be positioned in either cavity as this occurs, and that once Lipid XX has been synthesized, the TP reaction can take place. Cryo-EM, MD, and DEER determine that PBP2 is dynamic and flexible, which could

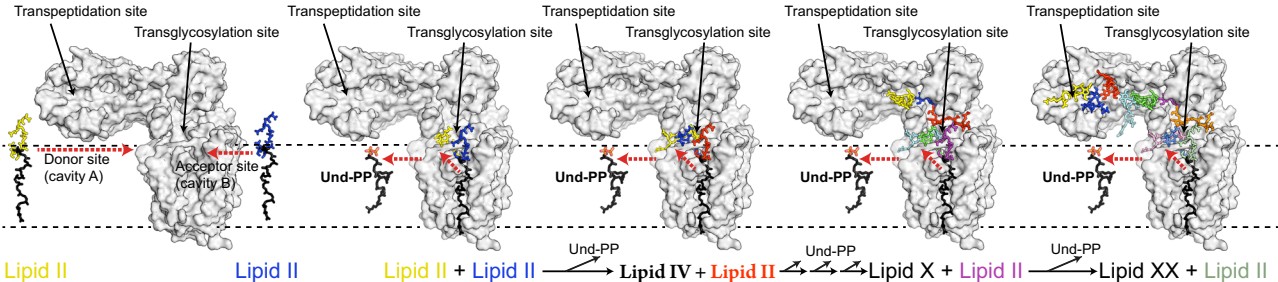

**Fig. 6 | Elongation of the glycan strand.** Modelling of the different steps that occur during elongation of the glycan strand. Starting from the left RodA-PBP2 is shown in complex with two Lipid II molecules. Once these two molecules are connected through the GT activity of RodA, Und-PP is released from cavity A either between TM helix 7 and the core of RodA, or under the JM helices into the membrane. We have modelled RodA-PBP2 bound sequentially to Lipid II, Lipid IV, up to Lipid XXII, and here we show different complexes during the elongation until the glycan chain reaches the active site of PBP2. RodA-PBP2 is shown as surface representation in grey. The previously suggested opening and closing of the head and anchor domain of PBP2 upon binding of the regulator protein MreC is not modelled here[15,23,24,87]. The Und-PP lipid is represented in black sticks and the disaccharides with the pentapeptide stem attached as sticks in different colors.

allow for the accommodation of the growing glycan strand and facilitate its attachment to the pre-existing PG.

RodA has structural similarities with a number of distinct GTs[26], including a ligand binding cavity created by a TM helix protruding away from the main helical bundle and a short amphipathic helix which lies parallel to the membrane, containing a highly conserved arginine residue, which provides coordination to the Und-PP product of the reaction[26]. Proteins which retain this functional motif for Und-PP binding include WaaL O-antigen ligase[26], and other members of the GT-C family of Und-PP dependent transferase and polymerases for which there until recently was poor structural definition[27]. Sequence conservation strongly indicates that this motif is retained across species. We propose that the mechanism described here for RodA is conserved across other SEDS proteins, including *B. subtilis* SpoVE that serves as in vivo system for the present study.

In summary, using an integrated approach centered around the cryo-EM structure of the *E. coli* RodA-PBP2 complex in the close to native environment of a nanodisc, combined with biochemical assays, genetic analysis, MD simulations and DEER experiments, we have generated a model for how this processive machinery functions to synthesize PG. This work will facilitate design structure-based inhibitors for the essential GT activity of RodA and other SEDS Lipid II polymerases for which none are known.

## Methods

### SEDS-PBP fusion design
Using the *E. coli* FtsW sequence as a seed, 189 different bacterial SEDS protein orthologs were identified by homology. These orthologs were cloned in a high throughput pipeline with a decahistidine tag at the N or C terminus separated from the target gene by a TEV cleavage site by NYCOMPS/COMPPÅ scientists (New York Consortium on Membrane Protein Structure/Center on Membrane Protein Production and Analysis) housed at the New York Structural Biology Center (NYSBC). These targets were screened for expression and monodispersity by size exclusion chromatography in a variety of detergents as previously described[19,39,40]. Five resulting SEDS targets that were most favorable for structural characterization were used to design SEDS-PBP fusion proteins, based on the *Bacillus subtilis* SpoVE-SpoVD and *E. coli* FtsW-PBP3 fusions genetically and functionally characterized previously[13]. The PBP partners of the SEDS proteins were identified, and PCR amplified from genomic DNA (provided by NYCOMPS/COMPPÅ) from the following bacterial strains: *Enterococcus faecalis* V583, *Klebsiella pneumoniae subsp. pneumoniae* MGH 78578, *Escherichia coli* str. K-12 substr. MG1655, *Streptococcus pneumoniae* TIGR4, and *Escherichia fergusonii* ATCC 35469. PBPs were inserted by Gibson assembly[41,42] at the C-terminal end of SEDS genes with a TSGSGSGS linker between the SEDS gene and the PBP gene (see Supplementary Table 4). Construct design is illustrated in

Supplementary Fig. 1a. All resulting clones were sequence verified by Sanger Sequencing (Macrogen). UniProt IDs for SEDS proteins and PBP proteins of the eight unique fusion proteins are given below in Supplementary Table 1. The *Escherichia coli* fusion (RodA UniProt ID 0ABG7; PBP2 UniProt ID P0AD65) in the pNYCOMPS-N23 vector resulted in the best expression levels and was carried forward for structural studies by cryo-EM.

### Protein expression, purification in detergent, and reconstitution in nanodisc
The RodA-PBP2 fusion from *E. coli* in the pNYCOMPS-N23 vector was used to transform 50 μL of BL21(DE3) pLysS *E. coli* competent cells, which were grown overnight in 20 mL of 2xYT media supplemented with 100 μg/mL ampicillin and 35 μg/mL chloramphenicol at 37 °C with shaking at 220 revolutions per minute (RPM). The next day 800 mL of 2xYT media was inoculated with 10 mL of starter culture and left to grow at 37 °C with shaking at 220 RPM until $OD_{600}$ was 0.8-1.0. Temperature was reduced to 22 °C and protein expression was induced with 0.2 mM isopropyl β-D-1-thiogalactopyranoside (IPTG), and the culture was incubated for 4 hours with shaking at 220 RPM. Cells were harvested by centrifugation at 3220 x g for 15 minutes at 4 °C, resuspended in 1x Phosphate Buffered Saline (PBS) and centrifuged again, then stored at −80 °C.

Cell pellets were thawed and resuspended in buffer containing 20 mM 4-(2-hydroxyethyl)-1-piperazineethanesulfonic acid (HEPES) pH 7.0, 200 mM NaCl, 20 mM $MgCl_2$, 4 μL/100 mL RNase, 10 μg/mL solid Dnase, 1:1000 cOmplete™, EDTA-free Protease Inhibitor Cocktail (Roche), 1 mM tris(2-carboxyethyl)phosphine (TCEP), and 1 mM phenylmethylsulfonyl fluoride (PMSF). A glass homogenizer was used to resuspend the cell pellets using 6 mL of buffer for each 1 gram of cell pellet mass. The resuspended cells were lysed by passing through an Emulsiflex C3 homogenizer (Avestin) at 15000 psi for 3 passages. Lysate was ultracentrifuged at 134000 x g for 30 minutes at 4 °C. Supernatant was discarded and membrane pellets were resuspended using a glass homogenizer in 30 mL High Salt wash buffer (20 mM HEPES, 500 mM NaCl, 10 μg/mL solid Dnase, 4 μL/100 mL RNase, 1 mM TCEP, 1:1000 cOmplete™, EDTA-free Protease Inhibitor Cocktail (Roche), 1 mM PMSF) per 800 mL of cell culture. Resuspended membranes were ultracentrifuged at 134000 x g for 30 minutes at 4 °C. Supernatant was discarded, and membrane pellets were resuspended and solubilized using a glass homogenizer and 30 mL buffer (20 mM HEPES pH 7, 500 mM NaCl, 20% glycerol, 10 μg/mL solid DNase, 4 μL/100 mL Rnase, 1:1000 cOmplete™, EDTA-free Protease Inhibitor Cocktail (Roche), 1 mM TCEP, and 1 mM PMSF and 1% n-Dodecyl β-D-maltoside (DDM; Anatrace) per 800 mL of cell culture. Solubilizing membranes were kept rotating for 2 hours at 4 °C. Subsequently, the sample was ultracentrifuged at 134000 x g for 30 minutes at 4 °C.

Supernatant is collected and combined with 500 μL of Ni-NTA agarose beads (Qiagen) for every 800 mL of cell culture that was harvested. The mixture rotates for 2 hours at 4 °C. The beads were then loaded onto a column and washed with 10 column volumes of buffer containing 20 mM HEPES pH 7, 500 mM NaCl, 60 mM imidazole, pH 7.5, 20% glycerol, and 0.1% DDM. Protein was eluted with 3 column volumes of buffer containing 20 mM HEPES pH 7, 500 mM NaCl, 300 mM imidazole pH 7.5, 20% glycerol, and 0.1% DDM. Imidazole was removed from the eluted protein by exchanging buffer to 20 mM HEPES pH 7, 500 mM NaCl, 20% glycerol, 0.1% DDM and 1 mM TCEP using a PD-10 desalting column (Cytiva). The concentration of RodA-PBP2 fusion protein was measured using a nanodrop spectrophotometer (Thermo Fisher). RodA-PBP2 fusion protein was combined with 1-Palmitoyl-2-oleoyl-sn-glycero-3-phosphor-racglycerol (POPG) and Membrane Scaffold Protein 1E3D1 in the molar ratios of 1:300:5 respectively and rotated for 2 hours at 4 °C. Bio-beads (Bio-Rad) were added to the mixture, which was left to rotate at 4 °C overnight. Bio-beads were removed, and the mixture was combined with Ni-NTA agarose beads (same volume as used previously). The mixture rotated for 2 hours at 4 °C. The Ni-NTA beads were then loaded onto a column and washed with 10 column volumes of buffer containing 20 mM HEPES pH 7, 500 mM NaCl, and 60 mM imidazole. Reconstituted nanodiscs were eluted by adding 3 column volumes of buffer containing 20 mM HEPES pH 7, 500 mM NaCl, 300 mM imidazole. Eluted sample is concentrated to 500 μL using a 100 kDa MWCO centrifugal filter (Amicon). The nanodisc complex was further purified by loading onto a Superdex 200 increase 10/300 size-exclusion column (Cytiva) with a filtered and degassed buffer containing 20 mM HEPES pH 7, 150 mM NaCl and 1 mM TCEP.

### Single-particle cryo-EM vitrification and data acquisition

Purified RodA-PBP2 complex was concentrated to 0.66 mg/ml (5.9 μM) using a 100-kDa concentrator (Amicon). The sample was frozen using a Vitrobot (Thermo Fisher) by adding 3 μL of the purified protein complex to previously plasma cleaned (Gatan Solarus) 0.6/1-μm holey gold grid (Quantifoil UltraAuFoil) and blotted using 595 filter paper (Ted Pella, Inc) for 7.5 s with a blot force of 3 and a wait time of 30 sec at 4 °C with >95% humidity. Images were recorded using a Titan Krios electron microscope (FEI), at the Columbia University Cryo-Electron Microscopy Center, equipped with an energy filter and a K3 direct electron detection filter camera (Gatan K3-BioQuantum) using a 0.83 Å pixel size. An energy filter slit width of 20 eV was used during the collection and was aligned automatically every hour using Leginon[43]. Data collection was performed using a dose of ~58.5 e-/Å$^2$ across 50 frames (50 ms per frame) at a dose rate of approximate 16.1 e−/pix/s, using a set defocus range of -1 μm to -2.5 μm. A 100 μm objective aperture was used. 11,120 micrographs were recorded over a two-day collection.

### Cryo-EM Data processing

Movie frames were aligned using Patch Motion Correction implemented in cryoSPARC v.2.12[44] using a B-factor during alignment of 500. Contrast transfer function (CTF) estimation was performed using Patch CTF as implemented in cryoSPARC v.2.12. CrYOLO[45] was used to pick particles. Particle picking resulted in 3,147,165 particles which were then extracted in cryoSPARC with a 400 pixel box size binned 4 times. The particles were classified using 2D classification in cryoSPARC v.3.2 using a batch size per class of 400 and "Force Max over poses/shifts" turned off with 40 online-EM iterations and one full iteration. 2D classes with well-defined high-resolution features were selected resulting in a particle stack of 441,319 particles. The particles were then re-extracted using a 400 pixel box size without binning. One round of ab initio reconstruction was performed in cryoSPARC v.3.2 using three classes, with a maximum resolution set at 4 Å and an initial resolution at 9 Å. This resulted in two classes that were mirror images and one class with a shortened PBP2. We then went back to the binned

particle stack of 3,147,165 particles and ran heterogenous refinement in cryoSPARC v3.2 using the three classes from the ab initio reconstruction and two decoy classes and a "Batch size per class" of 30,000. This heterogenous refinement was run three times using the particles from the three ab initio classes from the previous heterogenous refinement as input for the next heterogenous refinement. From this final heterogenous refinement the top two classes were selected (927,369 particles) and the particles were re-extracted without binning. This was followed by a non-uniform refinement with a final resolution of 3.24 Å. The particles were further sorted by two class heterogenous refinement in cryoSPARC v3.2 using the map from the 3.24 Å non-uniform refinement and a 10 Å lowpass filtered map as input. We used a "Batch size per class" of 30,000 and an initial resolution of 5 Å and a "Resolution of convergence criteria" of 100. This was followed by a non-uniform refinement with a final resolution of 3.17 Å (600,855 particles). To achieve higher resolution for the TM region, the particles were further separated by a 3D variability analysis with a mask around the TM region of the protein excluding the nanodisc and a filter resolution of 4.5 Å. The particles were clustered into five clusters and three of these clusters (399,759 particles) were used as input for a non-uniform refinement with a final resolution of 3.10 Å followed by a local refinement with a mask around the TM region of the molecule to a final resolution of 2.97 Å for the TM region. To achieve higher resolution for the periplasmic part of PBP2 we used the 600,855 particles from the previous non-uniform refinement and ran 3D variability analysis with a mask around the periplasmic part of PBP2. The particles were clustered into ten clusters and the six best clusters were used as input for non-uniform refinement (236,435 particles) to a final resolution of 3.23 Å. This was followed by a beam tilt refinement by image shift groups, followed by another non-uniform refinement to a final resolution of 3.08 Å. We further sorted the particles with another round of 3D classification in cryoSPARC using four classes, target resolution of 4 Å and 10,000 particles per epochs online expectation maximization (O-EM). This identified a stack of particles which we further refined using non-uniform refinement to a final resolution of 3.14 Å. Finally a local refinement was performed using a mask around the periplasmic region to a final resolution of 2.95 Å.

### 3D variability analysis for PBP2 movement

The 3D variability analysis shown in Fig. 5a is based on a subset of particles where a local refinement with a mask around the transmembrane region was imposed. From this map and with a mask around the periplasmic part of PBP2 a 3D variability was performed, the particles were divided into 6 clusters.

### Structural model building and refinement

An initial model of RodA and the TM helix of PBP2 was built as a homology model to the published RodA-PBP2 complex X-ray crystal structure[17]. The model was fitted to the map as a rigid body in Chimera[46], the model was subsequently adjusted to the current density using Namdinator[21] and further refined using Coot[47–49] and PHENIX[50,51] iteratively. For the soluble part of PBP2 the previously published X-ray structure of the soluble part of PBP2[22] was used as input in Namdinator[21] and further refined using Coot[47–49] and PHENIX[50,51] iteratively.

### Model analysis

A cavity search using the Solvent Extractor from Voss Volume Voxelator server[52] was performed using an outer-probe radius of 10 Å and inner-probe radius of 2 Å. Chimera[46], PyMOL and ChimeraX[53] were used to visualize the structures in the figures.

### Lipid II preparation

Dansylated lysine version for gel visualization studies of GT activity (dansyl lysine Lipid II), was produced by in vitro recapitulation of the synthetic pathway as detailed previously[54].

## Polymerization of Lipid II by RodA-PBP2 and RodA alone

The glycosyltransferase activity by RodA-PBP2 was demonstrated using visualization of fluorescently labelled dansyl Lipid II molecules using a Tris-Tricine acrylamide gel based electrophoresis method[55]. Detergent solubilized RodA-PBP2 protein (1.5 µL) at a concentration of 2.7 µM in 300 mM imidazole, 250 mM NaCl, 20 mM HEPES, 0.05% n-Dodecyl-B-D-Maltoside (DDM), 20% Glycerol was added to 13.5 µL reaction buffer (10 mM MgCl$_2$, 100 mM NaCl, 50 mM HEPES, 20% DMSO, 0.03% Lauryldimethylamine oxide (LDAO), 10 µM dansyl lysine Lipid II) to a final protein concentration of 0.27 µM. The reaction mixture was incubated at 37 °C for 1 hr 30 min to allow for RodA dependent polymerization of Lipid II.

The resultant Lipid II polymer was denatured at 95 °C to stop the reaction and mixed with 5x loading dye (50 mM Tris-HCl pH 8.8, 4 % SDS, 40 % glycerol, 0.01 % bromophenol blue, DTT 200 mM) prior to electrophoresis on a Biorad Criterion 16.5 % gel run at 110 V for 80 mins with anode gel running buffer (0.1 M Tris-HCl pH 8.8) and cathode gel running buffer (0.1 M Tris-HCl pH 8.25, 0.1 M Tricine, 0.1% SDS). The gel was visualized by 10 second exposure to UV on a BioRad GelDoc imaging system using Biorad imaging capture software. Concentration dependent assays were performed in the same manner, with altered concentrations of RodA-PBP2 or Lipid II added. MTSL labelled sample was prepared as described in EPR sample preparation below, before assay.

## Bocillin labeling of PBP2

Fluorescent Bocillin™ labeling of PBP2 was performed by incubating purified 25 µL RodA-PBP2 fractions with 1 µL bocillin (1 mg/mL) for 15 min. The resulting mixtures were run on an 4-15% SDS page gel, then visualized for 4 s by the using a BioRad gel documentation system[56]. The images were visualized at maxima to include the fluorescein ladder, and no other marker, therefore only visualizing bocillin stained proteins on the gel.

## Statistics and Reproducibility

All gel lanes were repeated at least three times with similar results.

## Mutagenesis of pNYCOMPS-N23-RodAPBP2 *E.coli* for functional analysis

Mutagenesis of pNYCOMPS-N23 RodA-PBP2 was performed using the QuickChange II Site-Directed Mutagenesis Kit (Agilent) with custom primers (see Supplementary Table 4).

## Conservation and Co-evolution analysis

A multiple sequence alignment for RodA was generated using MMseqs2[57] by searching both UniRef100 and environmental sequence sets for homologous proteins. Weblogo3[58] was used to analyze and represent the conservation of the aligned sequences of RodA as a Weblogo, with the *E. coli* K12 W3110 sequence used as reference. Consurf was used to represent the amino acid conservation on the surface of RodA[59].

Co-evolution analysis was performed using GREMLIN[60]. The *E.coli* K12 W3110 sequences of RodA (*mrdb*) and PBP2 (*mrdA*) were used as input sequences to identify homologues with a cut-off of E$^{-10}$. Based on this paired sequence alignment, GREMLIN was used, with default parameters to find the co-evolutionary contacts within either RodA or PBP2 and also between RodA and PBP2.

## Mutagenesis, spin labeling and sample preparation for EPR

Two native cysteines in *E. coli* RodA-PBP2 fusion (pNYCOMPS-N23 vector) were mutated to glycine and alanine (Cys82Gly and Cys133Ala) and cysteines were then introduced to the resulting cysteine-free background for site-directed spin labelling (SDSL) at residues hypothesized to create informative distance distributions

by DEER EPR spectroscopy based on the available structural information. All mutagenesis was performed using the QuickChange Lightning Site-Directed Mutagenesis Kit (Agilent) or QuickChange II Site-Directed Mutagenesis Kit (Agilent) with custom primers (see Supplementary Table 4). All mutants were tested for enzymatic function and bocillin binding as described above. In preparation for spin labelling, RodA-PBP2 mutants were purified in detergent as described above but were exchanged to a non-reducing buffer containing 20 mM HEPES pH 7, 500 mM NaCl, 20% glycerol, 0.1% DDM using a PD-10 desalting column (Cytiva) after elution. S-(2, 2, 5, 5-tetramethyl-2,5-dihydro-1H-pyrrol-3-yl)methyl methanesulfonothiolate (MTSL; Santa Cruz Biotechnology) was added at a 15:1 MTSL to protein molar ratio. The reaction was incubated at 4 °C overnight with agitation while protected from light. Excess spin label was removed with a PD-10 desalting column (Cytiva) using the same non-reducing buffer. The resulting protein was concentrated to 100-150 mM. For continuous-wave (CW) EPR experiments, 7 µL of concentrated protein was loaded into pyrex capillaries (0.6 mm id x 0.84 mm od; Vitrocom) and measured at room temperature. For pulsed (DEER) EPR experiments, deuterated glycerol was added to the concentrated protein sample to a final concentration of 20% (v/v) and 15-20 µL of sample was frozen in quartz capillary tubes (1.6 mm od x 1.1 mm id; Vitrocom) using a bath of dry ice and iso-propanol. Frozen samples were stored at -80 °C until pulsed EPR data were collected.

## Continuous-Wave and pulsed EPR data acquisition and analysis

CW EPR measurements were taken using an X-band Bruker EMX continuous wave spectrometer with an ER4123D dielectric resonator (Bruker Biospin) at room temperature. CW Spectra were baseline corrected and normalized using Lab-VIEW software (provided by C. Altenbach, University of California at Los Angeles). Pulsed (DEER) EPR measurements were taken at 50 K with a Q-band Bruker E580 EPR Spectrometer (Bruker Biospin) equipped with a 300 Watt traveling wave tube amplifier (Applied Systems Engineering) and an EN5107D2 resonator. A standard four-pulse DEER sequence was used for all measurements with p/2 and p pulse lengths varying with sample. A pump frequency is set at the maximum of the nitroxide spectrum and the observed frequency is set to 75 MHz lower. Increasing inter-pulse delays at 16 ns increments were utilized with a 16-step phase cycle during data collection. Accumulation times were typically between 12 and 36 hours, with a dipolar evolution time between 3 and 3.5 ms. Dipolar evolution data were processed using DEERAnalysis[61] with Gaussian model fitting or the DEERNet neural network plugin[62]. MTSL rotamers were attached to structures obtained from cryo-EM data or from MD simulations in silico with Multiscale Modeling of Macro-molecules (MMM)[63] using the default rotamer library. Nitroxide-to-nitroxide distances were calculated from resulting structures and binned to the closest angstrom for comparison with experimental data.

## In vivo studies in *Bacillus subtilis*

Assays were carried out as described in Fay et al.[13] Briefly, mutations were introduced into a SpoVE-SpoVD construct in a strain lacking *spoVD* and *spoVE* and sporulation (heat sensitivity) was assessed.

## Lipid II Ligand Docking

Autodock vina-carb[64] was used to dock Lipid II, Lipid II C55 mDAP, and peptidoglycan fragments to the identified cavities and crevices of the RodA-PBP2 complex. Docked poses were converted to CHARMM36m and energy minimized to optimize the binding orientations and to ligate docked fragments of peptidoglycan to allow the formation of the longer polymerized lipids, e.g., Lipid XX, with parameters developed based on those previously published[65].

## Molecular dynamics simulations

**Coarse grained simulations.** All coarse-grained (CG) MD simulations used the Martini 3 forcefield[66,67]. Martini 3 topologies with elastic networks were generated for RodA and PBP2 protein chains using Martinize2[66]. The DSSP program was used for secondary structure assignment[68], and intra-chain elastic network force constants were set to 500 kJ mol$^{-1}$ nm$^{-2}$ with upper and lower elastic bond cut-offs of 1.0 and 0.5 nm, respectively.

Martini 3 bead types and mapping to the Lipid II molecule were performed manually, converting from previously published Martini 2 Lipid II parameters[69]. Changes were made according to the amino acid beading in Martini 3[67] and the suggested bead types from the protocol for creating small molecules in Martini 3[70]. To refine the parameters for Lipid II, atomistic simulations (3 ×100 ns) were performed using previously published parameters[65] obtained from the authors in a PE:PG (3:1) membrane with a single copy of Lipid II. CG simulations were also performed with the same composition (5 ×1 μs). Representations of CG and atomistic Lipid II molecules are shown in Supplementary Fig. 5d. Distributions of distances and angles were measured using gmx tools distance and gangle. For the all-atom simulations, the atoms were grouped according to their bead types and the center of geometry measured. The solvent surface accessible area was measured using the gmx tool sasa. Plots were created using Matplotlib[71].

All model-systems were prepared in 3D periodic boxes by using the Memembed[72] and insane[73] methods to generate protein-solvating symmetric PE/PG bilayers (4:1 ratio of PE:PG). Where applicable, two molecules of Lipid II were positioned at random positions in the upper (periplasmic) leaflet. Initial box dimensions were set to 5.0 nm plus the maximum protein diameter, approximately 11x11x11 nm for RodA and 13x14x15 nm for RodA-PBP2 systems. Using insane, remaining voids were filled with water beads, with sodium and chloride ions at placed random positions representing a neutralizing salt concentration of 0.15 M. Systems were then subjected to steepest descent minimization with a tolerance of $F_{max} = 100$ kJ mol$^{-1}$ nm$^{-2}$. In total, the RodA and RodA-PBP2 systems consisted of approximately 11,000 and 24,000 beads, respectively.

All production CG simulations were conducted using GROMACS 2021[74] using the built-in leap-frog integrator with a timestep of 0.02 ps unless otherwise stated. All production simulations sampled isothermic-isobaric ensembles at 310 K using the V-rescale thermostat ($\tau_t = 1.0$)[75] and the C-rescale barostat for semi-isotropic pressure coupling at 1.0 bar ($\tau_p = 12.0$)[76]. Pre-production equilibration runs used the Berendsen barostat and a timestep of 0.01 ps. Separate coupling groups were used for protein, lipid and solvent molecules (i.e., waters and ions). For electrostatics, the reaction-field method was used with a Coulomb cut-off of 1.1 nm ($\varepsilon_r = 15$ and $\varepsilon_r = \infty$ for r > 1.1 nm), and van der Waals (VdW) interactions also used a cut-off of 1.1 nm (both with the Verlet cut-off scheme). The P-LINCS algorithm expanded up to 4$^{th}$ order was used for the treatment of holonomic constraints[77]. Each system was equilibrated for 10 ns, after which 10 μs production runs were prepared from the coordinates and velocities of the final frames of the equilibration trajectories. For RodA and RodA-PBP2, 50 repeats of the production simulations were conducted.

Density maps for the MD simulation in Fig. 3d were prepared by concatenating all repeat trajectories centered on RodA, then least-squares fitting to RodA backbone beads (using MDanalysis[78]) and using the open-source, community-developed PLUMED library[79] to log the x and y coordinates of the geometric centers of Lipid II phosphate beads. Because of the standard orientation of the protein in the bilayer (from Memembed), the z-axis of the simulation box is always perpendicular to the membrane plane and the resulting histogram of the x and y coordinates is functionally a bilayer-facing 2D projection of the particle density. The seaborn API was used to perform the kernel density estimate presented in the final figures. Lipid II interaction analysis was performed using PyLipID[25] to identify binding sites, interacting

residues and occupancy times for the bound Lipid II molecules from the CG simulations.

**Atomistic simulations.** Atomistic simulations for apo RodA and RodA-PBP2 were set up using the same pipeline as CG simulations, except using the Martini 2.2 forcefield[66] and position restraints for protein backbone beads with force constants set to 1000 kJ mol$^{-1}$ nm$^{-2}$. After 50 ns of CG equilibration the CG systems were converted into atomistic systems using the CG2AT program with the built-in "align" method to direct protein geometry towards the prepared EM structure (pre-CG-relaxation) during the conversion[80]. Alignment suggested a typical RMSD of only 0.14 nm over the course of CG equilibration. Overall, this protocol enabled preparation of all-atom membrane protein systems in well-equilibrated bilayers.

Where relevant, Lipid II and its polymerized forms (i.e., Lipid IV, Lipid XX etc) were added to the atomistic systems post-conversion (i.e., before atomistic equilibration) by aligning and energy minimizing the docked conformations in the two binding cavities observed (A and B).

Production simulations consisted of three repeats of 500 ns, each continued from a distinct equilibration trajectory. The system representing the cryo-EM structure of RodA-PBP2 with bound Und-PP was simulated for three repeats of 1 μs each. All atomistic simulations used the GROMACS 2021 leap-frog integrator with a timestep of 0.002 ps, the CHARMM36m forcefield[81] and TIP3P water model[82,83]. Simulations were performed in the isothermal-isobaric ensemble at 310 K with the V-rescale thermostat ($\tau_t = 0.1$), and the Parrinello-Rahman barostat for semi-isotropic pressure coupling at 1.0 bar ($\tau_p = 1.0$)[84,85]. Particle-mesh Ewald (PME) was used for treatment of electrostatics[86], and VDW interactions used a (Verlet) cutoff of 1.2 nm. All bonds involving hydrogens were converted to holonomic constraints which were treated with the 4$^{th}$ order P-LINCS algorithm.

For the PBP2 dynamics heatmap in Fig. 4, each apo RodA-PBP2 production simulation was root-mean-squared fit to RodA in the first trajectory frame using MDAnalysis before calculation of simulated B-factors using the GROMACS utility gmx rmsf. The values presented in the final figure are from averaging over the five repeat simulations and have been mapped to the minimized RodA-PBP2 structure before equilibration.

## Lipid II polymerization video

A linear interpolation of Lipid II polymerization was created using GROMACS tool gmx morph. This movie illustrates the elongation of PG, possible conformational changes of RodA associated with polymerization and the potential transition of Lipid-linked products from Cavity B to Cavity A to allow the next Lipid II to bind.

## DFTB Cluster Model

From the atomistic MD simulations with Lipids-II-VI bound, trajectory frames were filtered based on simple distance cut-offs of 6.5 Å between D262, the site B lipid C4 hydroxyl (donor) moiety and the site A lipid C1 (acceptor) atom to highlight suitable initial geometries of the catalytically relevant atoms. From these, selected snapshots were used to construct cluster models for QM minimizations. QM atoms included a slab-shaped region of RodA surrounding D262 and the bound Lipid head-groups at each site. RodA atoms consisted of 6 separate whole peptide fragments spanning residues R48-K49, K97-W102, R109-Q111, D159-L160, E258-F263, and S340-G343. For each substrate molecule, two complete glycan units (GlcNAc-MurNAc) were included (i.e., Lipid II) but undecaprenyl lipid tails were truncated after the first isoprenyl fragment and similarly pentapeptide chains were truncated at the L-Ala Cα atom (inclusive of the sidechain). When defining the boundaries of the peptide fragments, only (nonpolar) bonds between main-chain carbon atoms were cut, so cuts were always made between Cα and amide (carbonyl) carbons at both ends (such that N-terminal ends of

the fragments always contain the carbonyl moiety from the preceding residue). All cut bonds were capped with added hydrogen atoms, and all boundary atoms were fixed in-place during optimizations (i.e., with frozen Cartesian coordinates). Overall, the cluster model consisted of 590 atoms in total, with an overall charge of -1 and singlet multiplicity. All QM calculations reported made used the ORCA program (version 5.0.3) and the GFN2-xTB semiempirical density-functional with ALPB implicit solvation (water). To obtain product structures from reactant structures, minimizations were steered with curated harmonic restraints extending the pyrophosphate leaving-group bond and pulling along the coordinate forming the glycosidic bond, after which the cluster geometry was re-relaxed without the restraints, converging on the nearest product minimum. The supplementary video is interpolated from a partially converged NEB calculation consisting of 8 Images connecting product and reactant geometries minimized from a representative snapshot.

### Reporting summary

Further information on research design is available in the Nature Portfolio Reporting Summary linked to this article.

## Data availability

The density maps have been deposited into the Electron Microscopy Data Bank (EMDB), with accession code EMD-41303, EMD-41304 and EMD-41299. The model has been deposited in the Protein Data Bank (PDB), with accession code 8TJ3. All data are available in the manuscript or the supplementary materials. Source data are provided with this paper in the source data file. Source data are provided with this paper.

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

## Acknowledgements

This work was supported by NIGMS grant R35GM132120 (To F.M.), R35GM141953 (J.D.), R35GM131829 (L.C), and F32GM136076 (M.B.D). DIR was funded by a Schaefer Research Scholars Program Award to Columbia University, Biological Sciences Research Council (BBSRC) Grant BB/N003241/1 and Medical Research Council (MRC) Grant MR/N002679/1. Research in PJS's lab is funded by Wellcome (208361/Z/17/Z), the MRC (MR/S009213/1) and BBSRC (BB/P01948X/1, BB/R002517/1 and BB/S003339/1). JJD's studentship was sponsored by the MRC (MR/N014294/1) and studentships for NSB and CLBG were sponsored by the BBSRC (BB/M01116X/1). CLBG also received funded from NERC grant NE/T014717/1 and the Antibiotic Research UK Charity. CMB's studentship is sponsored by the MRC (MR/N014294/1). This project made use of time on ARCHER and JADE granted via the UK High-End Computing Consortium for Biomolecular Simulation, HECBioSim (http://hecbiosim.ac.uk), supported by EPSRC (grant no. EP/R029407/1). PJS acknowledges Sulis at HPC Midlands + , which was funded by the EPSRC on grant EP/T022108/1, and the University of Warwick Scientific Computing Research Technology Platform for computational access.

## Author contributions

SEDS-PBP fusion design and cloning was undertaken by M.B.D with help from J.D. and B.K. R.N optimized expression of the E. coli RodA-PBP2 construct with help from K.A, S.C. and M.B.D. R.N purified protein for cryo-EM sample. R.N. collected and analysed the cryo-EM data. R.N. built the model with help from C.L.B.G and O.B.C. Gene editing for E. coli functional assays was performed by C.L.B.G. and J.P., C.L.B.G. and N.B performed protein expression and purification for E. coli functional assays. Functional in vitro Lipid II polymerization assays in E. coli were carried out by C.L.B.G. Gene editing for B. subtilis was performed by M.A.H and J.D. In vivo B. subtilis sporulation assays were conducted by M.A.H with supervision from J.D. All MD simulations were performed by P.J.S and J.D.C, with some scripts provided by O.V, parameters for Lipid II dynamics were designed by C.M.B. Autodock Lipid II binding experiments were performed by C.M.B. and C.L.B.G. The Lipid II substrate was synthesised by C.L.B.G and J.J.D. Co-evolution analysis was conducted by P.J.S and C.L.B.G. Gene editing for DEER experiments was done by M.B.D and C.L.B.G., and DEER data were collected and analysed by M.B.D. under supervision of L.C. R.N, C.L.B.G, M.B.D, B.B., J.D.C, P.J.S, L.C, J.D., F.M and D.I.R designed experiments and R.N, C.L.B.G, M.B.D, J.D.C, P.J.S, L.C, J.D., F.M and D.I.R wrote the paper. Oversight for the entire project was provided by F.M.

## Competing interests

The authors declare no competing interest.
