## [Peer Review File · Nature Communications]

Structural basis of peptidoglycan synthesis by E. coli
RodA-PBP2 complexREVIEWER COMMENTS

Reviewer #1 (Remarks to the Author):

The manuscript by Nygaard and colleagues reports the structure of *E. coli* RodA-PBP2 complex, and tries to address mechanistic questions on substrate binding, catalysis, and product elongation on RodA and on the crosslinking reaction on PBP2. Overall, the study is a thoughtful and quite substantial to address challenging questions in PG biosynthesis. The structural studies benefited from a systematic screening procedure that identified the most suitable targets, and the *in vitro* and *in vivo* functional studies are implemented by a highly experienced multidisciplinary team combining computation, EPR, microbiology, and biochemistry.

While I do not have major concerns on the experiments and the outcomes, structure determination or functional analysis, I am not convinced that the data are sufficient to support the current model on the movement of the product from the acceptor site to the donor site. I feel that it is necessary to obtain a structure of RodA-PBP2 in complex with a Lipid X. I also do not see compelling evidence that the hydrophobic grooves of Site A or Site B be assigned as binding sites for the acyl chains. How strong is the outcome of simulation? Could recognition of a substrate come mainly from the sugars while letting the acyl chain float in the membrane?

Overall I am in favor of the manuscript being published as is, with cautionary statements on the final model.

Reviewer #2 (Remarks to the Author):

Nygaard et al. present the cryo-EM structure of the RodA-PBP2 complex active in peptidoglycan synthesis during cell elongation of rod-shaped bacteria. They performed extensive mutagenesis studies to identify the binding sites for lipid II and the growing glycan chain and present a model in which a processive glycan polymerization by RodA places the peptides into the TP site of PBP2. These exciting data substantially enhance the knowledge of the mechanisms of peptidoglycan synthesis.

Main points:

1. I understand that the structure determination and activity assays were performed with a RodA-PBP2 fusion protein. Can the authors add statements to indicate whether the two activities of the complex are

likely interdependent or not? For example, based on their data, would RodA be active without the bound PBP2? Are residues of PBP2 close to the GTase active site of RodA or could PBP2-binding affect the GTase site, hence GTase activity? And would PBP2 be active without the delivery of nascent glycan chains from RodA?

2. They don't provide evidence for TP activity of PBP2. SI Fig. 1g-i shows mass spectrometry data, but the interpretation of these data is not correct. The expected cross-linked product after GTase and TPase reactions with mDap-type lipid II, followed by mutanolysin treatment, is a GlcNAc-MurNAc-tetrapeptide-pentapeptide-MurNAc-GlcNAc molecule, which has a neutral mass of 1931.81 amu (C₇₇H₁₂₅N₁₅O₄₂) consistent with the reference they provided (Catherwood et al. 2020: neutral mass 1931.82 amu). However, they assigned a mass of 1948 amu to the (protonated) ion which is 16 amu higher than the expected mass of the protonated ion of the TP product, 1932 amu. It is also worrying that the 1948 signal has a very low intensity in the ms spectra (SI Fig. 1h). The ms/ms fragmentation spectra (SI Fig. 1i) shows again the wrong amu of 1948 for the TP product, and the proposed fragment ion structures are completely wrong: The structure of the 1039 amu fragment shows 4 alkynes (triple CC bonds), which cannot be generated through fragmentation of the cross-linked TP product. The 652 amu fragment lacks a MurNAc residue, but has a different moiety at this position, it also has NH-C(OH) moieties instead of amide NH-CO bonds and, on the left side, there is a carbon that carries two hydroxyl groups, which does not make sense chemically (these hydroxyls would lose H₂O and convert to carbonyl group). The same problem (2 hydroxyls at one carbon) exists on the structure shown next to the 265 amu ion. In my view, the ms data they have used to prove TP activity are highly dubious. TP activity should be removed from the manuscript altogether because the presented data don't provide evidence for TP activity of PBP2.

3. GTase activity, e.g. Fig. 2e, 4b, SI Fig. 1,7,9: Can they quantify the amount of lipid II consumed? Do they understand from structural analysis why certain mutations affect the length distribution of the glycan chains produced?

4. They don't provide oligonucleotide sequences used for cloning and therefore the amino acid sequences of the RodA-PBP2 construct cannot be deduced by the readers. Hence, experiments cannot be repeated by others. Can you add a table of the oligonucleotides used for cloning and information about the amino acid sequences of the (fusion) constructs?

Minor points:

1. Have they tried to determine the identity of the "contaminant" shown in SI Fig. 1j?

2. Movement of the glycan chain towards the TP site: Does LipidXX has 10 disaccharide (GlcNAc-MurNAc) units? In the figures/cartoons (Fig. 1a it looks there are less than 10 disaccharide units needed to span from the GT to the TP sites. Can they indicate the number of disaccharides of a chain that spans from the GT site to the TP site?

3. Have they tried to inhibit the GT reaction with Moenomycin to exclude that some of the RodA-PBP2 complexes contain contaminating class A PBP activity?

4. Methods section: polymerization of Lipid II by RodA-PBP2: please provide protein concentrations in micro-molar. Also, here they wrote that dansyl lysine lipid II was used, but the main text and SI Fig. 1b say that dansyl mDap lipidII was used. Please clarify.

5. SI Methods, Mutagenesis: the sequences of the "custom primers" should be given.

Reviewer #3 (Remarks to the Author):

The manuscript by Nygaard and co-workers is a tour-de-force that includes the characterization of the RodA-PBP2 complex from *E. coli* using not only cryo-electron microscopy but also biochemistry, biophysics, MD simulations and in vivo approaches. The work is exciting and timely. It extends the structural characterization of the RodA-PBP2 complex from *T. thermophilus* published in 2020 and provides exciting insights into catalytic details and conformational modifications of the complex. However, a few points must still be addressed.

Lines 105-110: The choice of characterizing the PBP2-RodA using fused proteins should be described in greater detail. Did all constructs carry the same GSGSGS linker between PBP2 and RodA?

Lines 139-140: Supplementary Figure 4a does not have any information regarding residues 344-400 and 431-456. Can authors clarify on the figure which are the residues that cannot be traced in the map?

Lines 180-182: it would be useful to be able to visualize a model of Und-PP docked within the density.

Lines 191-199: the strategy of testing the R210A mutation in *B. subtilis* sporulation is interesting but curious; readers would expect it to be tested in *E. coli*. Was it not possible because transformants were not obtained? Why was this specific strategy chosen? This should be clarified.

Lines 196-197: authors state that '... GT activity is severely reduced when Arg48 is mutated to alanine (Fig. 2e and Supplementary Fig. 7a) ...'. Supplementary Fig.7a does not display any results regarding GT activity, only protein expression. In addition, this section of the text lies within a paragraph that highlights the *B. subtilis* work. Thus, when authors mention this mutation, readers also refer to Supplementary Table 3, where 50% of the mutants have in vivo sporulation results that are 'not determined' (including R48A). The right part of the table simply repeats what has already been shown in the main figures. In this reviewer's opinion, the sporulation data should be displayed and discussed more clearly, and the reason for the high amount of ND results should also be clarified.

Line 203: authors state that ‘... This loop is intrinsically flexible, as observed in our MD simulations (Supplementary Fig. 5b)’. For clarity, the loop should be somehow highlighted in the figure (there is no mention of it in Sup Fig 5b).

Lines 204-206: authors state that ‘... PL2 has several highly conserved residues, including Trp102 and Gln111, that are invariant in all species analyzed (Supplementary Fig. 6), and the positively charged residues Lys97 and Arg101.’ Despite the fact that it is evident from the figure that W102 and Q111 are highly conserved, this does not seem to be the case for Lys97 and Arg101. In addition, later on in the paragraph, authors state that Arg101 ‘is the most conserved of the two’, but again there is no indication regarding how authors calculated this.

Line 209 and elsewhere: it is not totally clear to this reviewer why Supplementary Figure 7a is indicated associated to statements related to activity; it seems to be a bocillin staining gel to confirm expression and protein folding.

Lines 223-226: the absence of relation between the in vivo and in vitro mutagenesis results calls into question the employment of *B. subtilis* as an in vivo model, as mentioned above.

Lines 339 and beyond: authors remind the reader that RodA and PBP2 constitute the core of the elongasome. That being the case, they should also discuss how their findings can be integrated into models of regulation of cell wall elongation that involve RodA, PBP2, and other proteins such as MreC and MreD. Can their work shed light on models proposed by Li et al (2020) *PLoS Genet* and Martins et al (2021) *Nat Comm*? These models should be discussed.

Still regarding the published models – Rohs et al (2018) *PLoS Genet*, Martins et al (2021) *Nat Comm* and Li et al (2020) *Plos Genet* all suggest that PBP2 activates RodA when bound to MreC, in a conformation where PBP2 head and anchor regions are apart. However, in the supplementary movie presented here, the reaction is suggested to occur with head and anchor in closed conformation. Despite the fact that this reviewer understands that the movie was generated using the coordinates obtained with their cryo-EM structure, authors should at least include this detail in their figure legend so that their findings can be appropriately put into a wider context.

Minor comments

Figure 2a: the choice of colors (green on green) is not the best to allow visualization of the cavities

Figure 5b: authors should clarify the meaning between the different ribbon thicknesses (one assumes that the thicker ribbon regions display a higher RMSF but this should be included in the figure legend)

Reviewer #4 (Remarks to the Author):

The manuscript by Nygaard et al is describing the structural and functional characterisation of *E. coli* RodA-PBP2 complex that is responsible for peptidoglycan biosynthesis. The cryo-EM structure revealed the architecture of the complex and mutagenesis was coupled to probe key residues in the activity of the complex. DEER measurements showed the existence of a different state compared to the cryo-EM structure.

Overall, the work lacks novel findings for publication in Nature Communications. Although they present the cryo-EM structure of the *E. coli* RodA-PBP2 complex and several biochemical data that point to a plausible mechanism for Lipid II polymerisation, I do not think the study provides any new insights that previous published work has not provided (<https://doi.org/10.1038/s41564-020-0687-z> and <https://doi.org/10.1038/nature25985>). The structures are very similar to the *Thermus thermophilus* ones and previous mutagenesis studies had identified key residues. If this study was in complex with lipid II then it would provide the novelty.

Major points:

1. The authors need to compare their structure to the previously published *T. thermophilus* complex. What are the differences between the two species? Why have they completely ignored previous work apart from a quick reference at the introduction?
2. Another concern is that all the assays have been performed in detergent and not a lipid environment. Why not reconstitute the complex in a liposome or at least use their nanodisc system to probe the activity of the WT and mutant proteins? Does that affect the amount of polymerisation?
3. The DEER measurements do not provide any new insights on the mechanism or dynamics of the protein. Firstly, it is performed in detergent and secondly in the absence of a substrate. Why not reconstitute the protein in nanodiscs and try apo and lipid II? Do the labelled proteins retain activity? How can the very short distance be interpreted? The analysis of the distances relative to the complex is not thorough or as it is written does not provide any mechanistic information. What's the difference in DEER measurement between Fig 3d and 5d? Why is there a single peak in the later?
4. The authors need to revise the structure as a clash score of 16 is not acceptable. If the high clash score comes from the PBP2 structure, why not use AlphaFold to generate a better model and dock it/model in the density.

5. They should also tune down the statement 'line 109:that its structure in a near native lipid environment would greatly facilitate a mechanistic' as their nanodiscs only contain POPG far from the Ecoli membrane composition of PE:PG:cardiolipin.

6. lines 379-381, need to be tuned down as I do not think the study provides insights on accommodating the growth of the glycol strand. The study has identified the residues important for polymerisation but there is no evidence of the process based on the current work.

Reviewer #5 (Remarks to the Author):

Nygaard et al. reported a cryoEM structure of the RodA-PBP2 complex. This complex synthesizes and elongates the peptidoglycan (PG), a bacterial cell wall structural component. The RodA is a glycosyltransferase (GT), which polymerizes the disaccharide of one Lipid II molecule with another disaccharide or oligosaccharide lipid molecule. The Lipid II molecule is a C55 pyrophosphate-linked disaccharide with a pentapeptide attachment. Therefore the PG chain elongates by two carbohydrates after each linking reaction is completed. The PBP2 crosslinks the pentapeptides attached to a freshly synthesized polymeric glycan and the existing PG. Inhibiting PG synthesis is a useful mechanism to prevent bacterial growth. Therefore, this work is highly relevant to the microbiology community.

The crystal structure of the RodA-PBP2 complex was already available. In this article, the authors proposed Lipid II binding sites and a mechanism for Lipid II polymerization. The authors proposed two cavities in the RodA structure: the donor cavity and the acceptor cavity. The donor cavity binds to the disaccharide while the acceptor cavity house the nascent intermediate PG as it elongates. I think the definition of these cavities is not well-defined throughout the manuscript. For example, the authors analyzed the conformational flexibility of RodA and focused on "TM helices 1-2 and 8-10 on one side, and the helical bundle of TM helices 3-7 on the other". However, in previous sections, cavity A is defined differently. "Cavity A is located between TM helices 6, 7 and 9, and is framed on one side by PH1 and on the other by TM helices 5 and 6 and the periplasmic loop (PL3) connecting the two" Cavity B was defined as "between TM helices 2, 3, 4 and 10". Therefore, the conformation analysis lacks clarity. The authors should make the connection and keep consistent definitions.

The authors showed that cavity B has higher particle density for Lipid II than cavity A over the total simulation time. Therefore, cavity B must be the acceptor site. However, additional high density was obtained near the TM1, which is stronger than the density at cavity A. The authors should elaborate and explain. Additionally, it is not clear what happened to the Und-PP after the reactions. Its fate is not explicitly clarified in the proposed mechanism and the model presented by the authors.

The authors showed that the GT activity of the W102F mutant remains similar. Is the aromatic nature of both amino acids necessary in this case? If so, the W102Y mutant will probably show similar effects. The

authors should discuss further the implication of the Trp102 mutations from a mechanistic point of view. Additionally, the authors discuss many mutations in the “RodA active site and mechanism of catalysis” section. It would be insightful to see MD simulations of those mutants and analysis.

The authors presented activity data for many GT mutants in Figure 2e and SI fig 7a. However, mechanistic insight is lacking for these. The authors may try to perform MD simulations of these mutants to gain atomistic insight into the mechanism since that is the manuscript’s focus.

The length of the simulations may be inadequate to capture the conformational dynamics relevant to the overall reaction. As pointed out by the authors, the MD simulations did not capture the 10-degree vertical tilt of PBP2 with respect to the bilayer. The MD simulations only show that they are “dynamic.” I would suggest longer MD simulations and enhanced MD simulations for better conformational sampling.

The authors stated, “We suggest that Arg210 in cavity A and Arg48 in cavity B, as well as the arginine residues in PL2 (97-111) (Arg101 and Arg109), could facilitate this mechanism through coordination of the phosphate head groups.” ☐ The authors should provide more quantifiable metrics and details.

I am not clear on how the DFTB calculations provide any insight. In the calculations, the authors connected two states by the Nudged Elastic Band method. However, this does not confirm the feasibility of the reaction. The authors should discuss the energy along the optimized NEB path to ensure feasibility. The movie does not provide enough information either.

Minor:

1. Definite of PL2 is missing in SI Fig 5b
2. Reference 5b and 5c is switched in the main text in line 310 and 319
3. It is very difficult to distinguish two conformations from Figure 5a.
4. Figure 6 could be annotated better with relevant information. For example, what is the groove the authors referred to in the main text?
5. The authors stated, “the tight coordination of the polyprenyl tails in both cavities A and B anchors the nascent PG to the membrane (Supplementary Fig. 10a).” This is unclear in Figure 10a. It only shows the growth of the PG and RMSF. The “tight coordination” is missing. Similarly, Figure 10b does not show the

secondary structure analysis, although the text claimed, “secondary structure of RodA-PBP2 is stable during the simulations (Supplementary Fig. 10b).”

Response to Reviewers

Nygaard et al. “Structural basis of peptidoglycan synthesis by *E. coli* RodA-PBP2 complex”

We are thankful for the overall positive and constructive feedback from the five Reviewers. Here we provide a point-by-point response to their comments.

Reviewer #1:

The manuscript by Nygaard and colleagues reports the structure of E. coli RodA-PBP2 complex, and tries to address mechanistic questions on substrate binding, catalysis, and product elongation on RodA and on the crosslinking reaction on PBP2. Overall, the study is a thoughtful and quite substantial to address challenging questions in PG biosynthesis. The structural studies benefited from a systematic screening procedure that identified the most suitable targets, and the in vitro and in vivo functional studies are implemented by a highly experienced multidisciplinary team combining computation, EPR, microbiology, and biochemistry.

We are grateful to the Reviewer for their appreciation of our work.

While I do not have major concerns on the experiments and the outcomes, structure determination or functional analysis, I am not convinced that the data are sufficient to support the current model on the movement of the product from the acceptor site to the donor site. I feel that it is necessary to obtain a structure of RodA-PBP2 in complex with a Lipid X. I also do not see compelling evidence that the hydrophobic grooves of Site A or Site B be assigned as binding sites for the acyl chains. How strong is the outcome of simulation? Could recognition of a substrate come mainly from the sugars while letting the acyl chain float in the membrane?

We thank the Reviewer for their comments. Both cavities are clearly defined in the MD simulations, both at coarse-grained representation and in our atomistic simulations. The cryo-EM structure also has density for a bound undecaprenyl pyrophosphate tail in cavity A. We agree that it remains uncertain exactly how the product of the reaction in cavity B moves to cavity A for the next round of the cycle. That said, this is a processive enzyme and we do provide a clear picture of how this process could plausibly proceed. The apposition of the two Lipid II molecules in both cavities positions the headgroups perfectly adjacent to the catalytic Asp262 to engage to permit transglycosylation.

Based on the Reviewer’s query we have further analysed our simulations for the interactions made Lipid II. This data very much shows that the interactions made between Lipid II and RodA are driven by contacts between arginine residues and the pyrophosphate group of Lipid II. This permits the sugars to engage with the periplasmic face of RodA. Importantly, the Reviewer is correct, this allows the polyprenyl tails to remain engaged with the membrane and within the cervices formed by the proteinaceous surface of RodA. In our atomistic simulations the polyprenyl tail is stably bound but interacts with both protein and lipid to anchor it to the membrane. We have updated the text to capture this.

Overall I am in favor of the manuscript being published as is, with cautionary statements on the final model.

We have changed the language in the discussion to be more speculative and less definitive in terms of our overall conclusion of the model. We appreciate that the Reviewer considers our manuscript suitable for publication.

Reviewer #2:

Nygaard et al. present the cryo-EM structure of the RodA-PBP2 complex active in peptidoglycan synthesis during cell elongation of rod-shaped bacteria. They performed extensive mutagenesis studies to identify the binding sites for lipid II and the growing glycan chain and present a model in which a processive glycan polymerization by RodA places the peptides into the TP site of PBP2.

These exciting data substantially enhance the knowledge of the mechanisms of peptidoglycan synthesis.

We pleased that the Reviewer finds our data exciting, and we have addressed their concerns below.

Main points:

1. I understand that the structure determination and activity assays were performed with a RodA-PBP2 fusion protein. Can the authors add statements to indicate whether the two activities of the complex are likely interdependent or not? For example, based on their data, would RodA be active without the bound PBP2?

Whilst RodA has been shown *in vitro* to have residual activity in isolation (Sjodt, *et al.*, *Nature*, 2018) subsequent papers suggest that the presence of the PBP TM helix induces a conformational change in RodA that significantly stimulates GT activity (Sjodt, *et al.*, *Nature Microbiology*, 2020). In support of this, the comparison of the published *T. thermophilus* RodA structure in isolation and in complex with PBP2 shows a large conformational change between the two RodA molecules.

We find a similar pattern in the *E. coli* system analyzed here. Indeed, RodA in isolation has very low activity, but a version of the fusion which terminates just after the transmembrane helix of PBP2 has significantly enhanced Lipid II polymerisation activity, which is comparable to the full length RodA-PBP2 product. Therefore, in our opinion we can reliably conclude that the presence of the single transmembrane helix of PBP2 in complex with RodA stimulates lipid II GT activity. These new data are included in the revised manuscript in Supplementary Fig. 1g-h.

Are residues of PBP2 close to the GTase active site of RodA or could PBP2-binding affect the GTase site, hence GTase activity?

The single TM helix of PBP2 forms tight hydrophobic interactions between TM helices 8 and 9 of RodA, and the association extends to part of periplasmic loop 2 (PL2). Conservation amongst interacting residues is high in both proteins and is particularly so towards the periplasmic end of the TMs and the loop region. There is also a region of interacting residues in the loop between PH1 and PH2 (RodA), and the linker domain of PBP2, but conservation is low here. As mentioned above, we are including new data to the revised manuscript showing that the addition of the PBP2 TM helix enhances Lipid II polymerisation *in vitro* compared to RodA alone.

And would PBP2 be active without the delivery of nascent glycan chains from RodA?

Whilst β -lactams drugs have been shown to bind to PBP2, we are not aware of prior work showing TP activity for PBP2 with substrate mimetics. This is an issue for the PBP field at large. In order to succeed in showing this, one would have to generate PG substrate mimetics of the TP donor and acceptor site. This is an extremely challenging task from a synthetic chemistry point of view and in our opinion beyond the scope of our current work.

*2. They don't provide evidence for TP activity of PBP2. SI Fig. 1g-i shows mass spectrometry data, but the interpretation of these data is not correct. The expected cross-linked product after GTase and TPase reactions with mDap-type lipid II, followed by mutanolysin treatment, is a GlcNAc-MurNAc-tetrapeptide-pentapeptide-MurNAc-GlcNAc molecule, which has a neutral mass of 1931.81 amu (C77H125N15O42) consistent with the reference they provided (Catherwood *et al.* 2020: neutral mass 1931.82 amu). However, they assigned a mass of 1948 amu to the (protonated) ion which is 16 amu higher than the expected mass of the protonated ion of the TP product, 1932 amu. It is also worrying that the 1948 signal has a very low intensity in the ms spectra (SI Fig. 1h). The ms/ms fragmentation spectra (SI Fig. 1i) shows again the wrong amu of 1948 for the TP product, and the proposed fragment ion structures are completely wrong: The structure of the 1039 amu fragment shows 4 alkynes (triple CC bonds), which cannot be generated through fragmentation of the cross-linked TP product. The 652 amu fragment lacks a MurNAc residue, but has a different moiety at this position, it also has NH-C(OH) moieties instead of amide NH-CO bonds and, on the left side, there*

is a carbon that carries two hydroxyl groups, which does not make sense chemically (these hydroxyls would lose H₂O and convert to carbonyl group). The same problem (2 hydroxyls at one carbon) exists on the structure shown next to the 265 amu ion. In my view, the ms data they have used to prove TP activity are highly dubious. TP activity should be removed from the manuscript altogether because the presented data don't provide evidence for TP activity of PBP2.

We thank the Reviewer for their expert analysis. Upon further inspection of the data, we agree that it may be best to remove from the revised manuscript the TP activity data presented initially and have proceeded accordingly.

3. GTase activity, e.g. Fig. 2e, 4b, SI Fig. 1,7,9: Can they quantify the amount of lipid II consumed? Do they understand from structural analysis why certain mutations affect the length distribution of the glycan chains produced?

We thank the Reviewer for these insightful questions. Whilst in theory it is possible to quantify the amount of fluorescent Lipid II consumed, in practice this is highly inaccurate, not least because the reaction does not proceed to completion under the conditions used. Quantification of the Lipid II polymers formed presents even more complex challenges. We therefore regard this PAGE based GT assay as qualitative, providing insight regarding GT activity and polymerisation characteristics. While acknowledging in full that quantitative GT assays are needed, our assay is, in our opinion, consistent with the current standard in the literature for other Lipid II GT enzymes and other lipid II labelling methods.

It is interesting to speculate on why some mutations lead to differences in the lengths of the polymers formed (usually shorter). One plausible interpretation is that this could be due to premature chain termination as observed in *S. aureus* PBP2 and MGT (Rebets, *et al.*, *ACS Chem Biol*, 2014). A quantitative methodology to assess either Lipid II consumed, or GT strand length is required to fully explain this. Our experiments indicate that mutations in some residues produce changes in GT activity in this regard, and we think it is important to bring this observation to the attention of the community.

4. They don't provide oligonucleotide sequences used for cloning and therefore the amino acid sequences of the RodA-PBP2 construct cannot be deduced by the readers. Hence, experiments cannot be repeated by others. Can you add a table of the oligonucleotides used for cloning and information about the amino acid sequences of the (fusion) constructs?

We have added a table with all the primers used in this study as Table 4, and we have also added the amino acid sequence in the same table.

Minor

points:

1. Have they tried to determine the identity of the "contaminant" shown in SI Fig. 1j?

Based on mass spec analysis we have determined the contaminant in the gel to be Cytochrome bo(3) ubiquinol oxidase subunit 2. We have replaced the gel with a more representative gel of the purified protein, where the contaminant is not as prominent.

2. Movement of the glycan chain towards the TP site: Does LipidXX has 10 disaccharide (GlcNAc-MurNAc) units?

Lipid XX has 10 disaccharides, and we have specified this in the text.

In the figures/cartoons (Fig. 1a it looks there are less than 10 disaccharide units needed to span from the GT to the TP sites.

We have updated Fig. 1a to reflect the number of disaccharides in our final model.

Can they indicate the number of disaccharides of a chain that spans from the GT site to the TP site?

Based on our modelling we find that, if bound to Cavity A, ten disaccharide units (Lipid XX) span from RodA to the TP site in PBP2. If bound to Cavity B, 11 disaccharide units are required (Lipid XXI). We anticipate that the updated Fig. 1a will clarify this.

3. Have they tried to inhibit the GT reaction with Moenomycin to exclude that some of the RodA-PBP2 complexes contain contaminating class A PBP activity?

We have assayed RodA-PBP2 in the presence of moenomycin and there appears to be no inhibitory effect, consistent with previous reports, and have added these data to Supplementary Fig. 1c in the revised version of the manuscript. Moreover, although class A PBPs are metal ion dependent and can be inhibited by EDTA, RodA-PBP2 cannot be inhibited by EDTA, suggestive of a distinct enzymatic mechanism.

4. Methods section: polymerization of Lipid II by RodAPBP2: please provide protein concentrations in micro-molar.

We have provided molar protein concentrations in the Methods section.

Also, here they wrote that dansyl lysine lipid II was used, but the main text and SI Fig. 1b say that dansyl mDap lipidII was used. Please clarify.

We apologize for the confusion. In response, we have modified Supplementary Fig. 1b and the main text to show and read, respectively, dansyl lysine lipid II as this was used in all the polymerization assays.

5. SI Methods, Mutagenesis: the sequences of the "custom primers" should be given.

We have added Table 4 with the sequences of the custom primers used in this study.

Reviewer #3:

The manuscript by Nygaard and co-workers is a tour-de-force that includes the characterization of the RodA-PBP2 complex from E. coli using not only cryo-electron microscopy but also biochemistry, biophysics, MD simulations and in vivo approaches. The work is exciting and timely. It extends the structural characterization of the RodA-PBP2 complex from T. thermophilus published in 2020 and provides exciting insights into catalytic details and conformational modifications of the complex. However, a few points must still be addressed.

We thank the Reviewer for these supportive comments.

Lines 105-110: The choice of characterizing the PBP2-RodA using fused proteins should be described in greater detail. Did all constructs carry the same GSGSGS linker between PBP2 and RodA?

The linker between the two *E. coli* proteins is TSGSGSGS, and this was the same for all the protein fusions used in the activity assays, structural studies and DEER experiments. We have edited the text to reflect this and added Table 4 showing the full sequence of the fusion construct.

Lines 139-140: Supplementary Figure 4a does not have any information regarding residues 344-400 and 431-456. Can authors clarify on the figure which are the residues that cannot be traced in the map?

We have highlighted these in Fig. 4a by adding green labels on the structure.

Lines 180-182: it would be useful to be able to visualize a model of Und-PP docked within the density.

We thank the Reviewer for this suggestion, which we have followed. We don't observe density for the entire lipid tail but were able to reliably fit in 16 carbon atoms from UndPP (See Fig. 2d).

Lines 191-199: the strategy of testing the R210A mutation in B. subtilis sporulation is interesting but curious; readers would expect it to be tested in E. coli. Was it not possible because transformants were not obtained? Why was this specific strategy chosen? This should be clarified.

We thank the Reviewer for their interest in this aspect of the study. It was not possible to generate an R210A mutant *in vivo* in *E. coli* since we do not have (and are not aware of) a genetic system in which RodA is under separate genetic control (from PBP2 and other genes). We did however mutate R210 in our *in vitro* assay, as shown in Fig. 2e. The *B. subtilis* genetic system has the advantage over *E. coli* in that we can quantitatively distinguish between complete loss of function (e.g., D262A/D263A) as compared to E258A/E259A which has a moderate defect. Such quantitative differences are not apparent in the *E. coli* system which is essentially qualitative.

Lines 196-197: authors state that ‘... GT activity is severely reduced when Arg48 is mutated to alanine (Fig. 2e and Supplementary Fig. 7a) ...’. Supplementary Fig.7a does not display any results regarding GT activity, only protein expression. In addition, this section of the text lies within a paragraph that highlights the B. subtilis work. Thus, when authors mention this mutation, readers also refer to Supplementary Table 3, where 50% of the mutants have in vivo sporulation results that are ‘not determined’ (including R48A). The right part of the table simply repeats what has already been shown in the main figures. In this reviewer’s opinion, the sporulation data should be displayed and discussed more clearly, and the reason for the high amount of ND results should also be clarified.

Supplementary Fig. 7a is provided as a control to give reassurance that any changes in activity seen in Fig. 2e are not due to lack of protein expressed or incorrect folding, as indicated. We have used this consistently with all of our mutants and refer to these bocillin assays whenever we show an *in vitro* assay activity gel. All our mutants have been purified on multiple independent occasions and tested independently to ensure that the data shown is consistent. In response to the Reviewer’s comment, we have modified the text to dissipate any source of confusion, and added data regarding the sporulation efficiency of mutants that demonstrated a defect in the biochemical assays (e.g., R48A, R101A/S103A, P257A/P258A, T261S/T262S) in order to confirm that the *in vitro* defect was also observed *in vivo*.

Line 203: authors state that ‘... This loop is intrinsically flexible, as observed in our MD simulations (Supplementary Fig. 5b)’. For clarity, the loop should be somehow highlighted in the figure (there is no mention of it in Sup Fig 5b).

We thank the Reviewer for this suggestion and in response have added a black box showing PL2 in Supplementary Fig. 5b.

Lines 204-206: authors state that ‘... PL2 has several highly conserved residues, including Trp102 and Gln111, that are invariant in all species analyzed (Supplementary Fig. 6), and the positively charged residues Lys97 and Arg101.’ Despite the fact that it is evident from the figure that W102 and Q111 are highly conserved, this does not seem to be the case for Lys97 and Arg101. In addition, later on in the paragraph, authors state that Arg101 ‘is the most conserved of the two’, but again there is no indication regarding how authors calculated this.

This was previously detailed in the methods, but we have expanded upon this, and are grateful to the Reviewer for having noticed this potential source of confusion. The *E. coli* RodA sequence was used with MMseqs2 to search for homologous sequences within both UniRef100 and the environmental sequence databases. This identified ~14,000 sequences. We then utilized Weblogo3 to evaluate and score each residue position for conservation. We have updated the text to improve this section of the manuscript.

Line 209 and elsewhere: it is not totally clear to this reviewer why Supplementary Figure 7a is indicated associated to statements related to activity; it seems to be a bocillin staining gel to confirm expression and protein folding.

As mentioned above, Supplementary figure 9a shows that the expressed and purified proteins are of the correct size and can bind bocillin, which is commonly used in the literature to provide proxy evidence supporting correct folding of the PBP. To the best of our knowledge, there is no equivalent assay for the SEDS proteins, beyond measuring GT activity.

Lines 223-226: the absence of relation between the in vivo and in vitro mutagenesis results calls into question the employment of B. subtilis as an in vivo model, as mentioned above.

The *in vivo* data reflects not just the GT activity (as the *in vitro* assay reports) but other aspects of SEDS protein function, such as interaction with other proteins, PBPs for example (e.g., Fay, *et al*, *Journal of Molecular Biology*, 2010 and Fraipont, *et al*, *Microbiology*, 2011) or its role in flipping the Lipid II precursor (e.g., Mohammadi, *et al*, *EMBO J*, 2011). In addition, interactions with the Lipid II would likely have different consequences for *in vitro* and *in vivo* function. So, as such, we would expect that although many (7/11) of the mutations affect both *in vitro* and *in vivo* function, there may be exceptions. In our opinion, understanding these differences awaits the establishment of additional *in vitro* assays for SEDS function, and is thus beyond the scope of this manuscript.

Lines 339 and beyond: authors remind the reader that RodA and PBP2 constitute the core of the elongasome. That being the case, they should also discuss how their findings can be integrated into models of regulation of cell wall elongation that involve RodA, PBP2, and other proteins such as MreC and MreD. Can their work shed light on models proposed by Li et al (2020) PLoS Genet and Martins et al (2021) Nat Comm? These models should be discussed.

We thank the Reviewer for their interest in the wider consequences of the data presented. At this stage, as we do not present any conclusive data on MreC or MreD, it seems overly speculative to propose a role for these additional components of the elongasome.

Still regarding the published models – Rohs et al (2018) PLoS Genet, Martins et al (2021) Nat Comm and Li et al (2020) Plos Genet all suggest that PBP2 activates RodA when bound to MreC, in a conformation where PBP2 head and anchor regions are apart. However, in the supplementary movie presented here, the reaction is suggested to occur with head and anchor in closed conformation. Despite the fact that this reviewer understands that the movie was generated using the coordinates obtained with their cryo-EM structure, authors should at least include this detail in their figure legend so that their findings can be appropriately put into a wider context.

We have updated the figure legend to add more details and references to the papers mentioned by the Reviewer, and are appreciative of this suggestion.

Minor *comments*

Figure 2a: the choice of colors (green on green) is not the best to allow visualization of the cavities

We have changed the color of the cavity volumes to orange creating a better contrast between the cavity volumes and the conservation rendered surface of RodA.

Figure 5b: authors should clarify the meaning between the different ribbon thicknesses (one assumes that the thicker ribbon regions display a higher RMSF but this should be included in the figure legend)

We have clarified this.

Reviewer #4:

The manuscript by Nygaard et al is describing the structural and functional characterisation of E. coli RodA-PBP2 complex that is responsible for peptidoglycan biosynthesis. The cryo-EM structure revealed the architecture of the complex and mutagenesis was coupled to probe key residues in the activity of the complex. DEER measurements showed the existence of a different state compared to the cryo-EM structure.

Overall, the work lacks novel findings for publication in Nature Communications. Although they present the cryo-EM structure of the E. coli RodA-PBP2 complex and several biochemical data that point to a plausible mechanism for Lipid II polymerisation, I do not think the study provides any new insights that previous published work has not provided (<https://doi.org/10.1038/s41564-020-0687-z> and <https://doi.org/10.1038/nature25985>). The structures are very similar to the Thermus thermophilus ones and previous mutagenesis studies had identified key residues. If this study was in complex with lipid II then it would provide the novelty.

We respect the opinion of the Reviewer, and we have addressed all the major and minor points raised below. The work of Sjødt et al 2018 and 2020 was indeed a landmark in the field of SEDS-PBP biology, but lacked insight on the GT reaction catalysed which we now address. In our revised manuscript we have been more exhaustive in comparing the published structures with our structure, and referenced this previous work more thoroughly. We believe, in agreement with one of the other Reviewers, that our manuscript “extends the structural characterization of the RodA-PBP2 complex from *T. thermophilus* published in 2020 and provides exciting insights into catalytic details and conformational modifications of the complex.”

Major points:

1. The authors need to compare their structure to the previously published T. thermophilus complex. What are the differences between the two species? Why have they completely ignored previous work apart from a quick reference at the introduction?

We appreciate the Reviewer for pointing this out. We have now added an additional figure comparing the two structures (Supplementary Fig. 4c), and a section in the results discussing their differences. Our work is independent of the previously published data, and in our opinion add novelty by focusing on an in-depth consideration of Lipid II binding, GT mechanism and its relationship to overall RodA-PBP2 function.

2. Another concern is that all the assays have been performed in detergent and not a lipid environment. Why not reconstitute the complex in a liposome or at least use their nanodisc system to probe the activity of the WT and mutant proteins? Does that affect the amount of polymerisation?

We have assayed RodA-PBP2 in nanodisc, and compared it to RodA-PBP2 in detergent – data presented in Supplementary Fig 1d – and found no detectable difference in activity. Purifying and reconstituting all mutants in nanodisc would have required a substantially larger amount of protein expressed and purified (increasing costs and time incommensurately), and since RodA-PBP2 expresses at relatively low levels we prioritized looking at a wider distribution of mutants in detergent instead of a select few after reconstitution in nanodiscs.

3. The DEER measurements do not provide any new insights on the mechanism or dynamics of the protein. Firstly, it is performed in detergent and secondly in the absence of a substrate. Why not reconstitute the protein in nanodiscs and try apo and lipid II? Do the labelled proteins retain activity? How can the very short distance be interpreted? The analysis of the distances relative to the complex is not thorough or as it is written does not provide any mechanistic information. What's the difference in DEER measurement between Fig 3d and 5d? Why is there a single peak in the later?

DEER measurements in Fig. 3 and 5 are of different sets of mutants that were designed to measure different motions of the complex – the dynamics within RodA and the dynamics of PBP2 with respect to RodA, respectively. We have added the specific mutations in the caption, which were referred to appropriately in the main text. These spin labeled mutants and an additional spin labeled mutant presented in this revised version of the manuscript were tested for activity (in detergent) and the results are shown in Supplementary Fig. 9.

Since the assays in detergent demonstrated that the mutants are functional and reconstitution in nanodiscs results in lower protein yields and therefore a much-reduced EPR signal, we relied on detergent reconstituted samples for a better – and more accurate – understanding of the

conformational dynamics of RodA-PBP2. The DEER data enhance our understanding of this complex beyond a single static state that is represented by the cryo-EM data and structure.

The additional mutant is included for comparison. A pair of spin labels was introduced into the PBP2 domain to demonstrate the domain itself doesn't have the conformational dynamics (single population with narrow distance distribution) observed between PBP2 and RodA (multiple populations and broad distance distributions), adding confidence and additional context in the interpretation of the other DEER data. The DEER data are not presented to elucidate a mechanism but should be taken to provide insight into the dynamics and conformational sampling that is occurring.

Finally, it is unclear which distance distribution the Reviewer is referring to in respect to an interpretation for shorter distances. However, the distances can be explained with several interpretations, and we are working on multiple additional pairs to triangulate a rational explanation for the conformations sampled. These additional experiments will be presented in a later manuscript and are out of the scope of the current work.

4. The authors need to revise the structure as a clash score of 16 is not acceptable. If the high clash score comes from the PBP2 structure, why not use Alphafold to generate a better model and dock it/model in the density.

We substantially improved the density of PBP2 (see Methods) and were able to improve the model and lower the clash score.

5. They should also tune down the statement 'line 109:that its structure in a near native lipid environment would greatly facilitate a mechanistic' as their nanodiscs only contain POPG far from the E. coli membrane composition of PE:PG:cardiolipin.

We have removed the words “near native”, so the sentence now reads “that its structure in a lipid environment would greatly facilitate a mechanistic”.

6. lines 379-381, need to be tuned down as I do not think the study provides insights on accommodating the growth of the glycol strand. The study has identified the residues important for polymerisation but there is no evidence of the process based on the current work.

We have toned down the statements about the growing glycan chain, to make it clear to the reader this is a model and not something the structure showed.

Reviewer #5:

Nygaard et al. reported a cryoEM structure of the RodA-PBP2 complex. This complex synthesizes and elongates the peptidoglycan (PG), a bacterial cell wall structural component. The RodA is a glycosyltransferase (GT), which polymerizes the disaccharide of one Lipid II molecule with another disaccharide or oligosaccharide lipid molecule. The Lipid II molecule is a C55 pyrophosphate-linked disaccharide with a pentapeptide attachment. Therefore the PG chain elongates by two carbohydrates after each linking reaction is completed. The PBP2 crosslinks the pentapeptides attached to a freshly synthesized polymeric glycan and the existing PG. Inhibiting PG synthesis is a useful mechanism to prevent bacterial growth. Therefore, this work is highly relevant to the microbiology community.

We thank the Reviewer for their positive assessment of our work.

The crystal structure of the RodA-PBP2 complex was already available. In this article, the authors proposed Lipid II binding sites and a mechanism for Lipid II polymerization. The authors proposed two cavities in the RodA structure: the donor cavity and the acceptor cavity. The donor cavity binds to the disaccharide while the acceptor cavity house the nascent intermediate PG as it elongates. I think the definition of these cavities is not well-defined throughout the manuscript. For example, the authors analyzed the conformational flexibility of RodA and focused on “TM helices 1-2 and 8-10 on one side, and the helical bundle of TM helices 3-7 on the other”. However, in previous sections,

cavity A is defined differently. “Cavity A is located between TM helices 6, 7 and 9, and is framed on one side by PH1 and on the other by TM helices 5 and 6 and the periplasmic loop (PL3) connecting the two” Cavity B was defined as “between TM helices 2, 3, 4 and 10”. Therefore, the conformational analysis lacks clarity. The authors should make the connection and keep consistent definitions.

We thank the Reviewer for highlighting this. We have further defined the two cavities in the manuscript. When we analyse the conformational flexibility of the two domains, we refer to movement of the substrate between the two cavities because of their relative conformational flexibility, not flexibility within the cavities themselves. For this reason, the helices defining the cavities are not equivalent to the helices in the two domains in the flexibility analysis. We hope this clarifies this source of confusion.

The authors showed that cavity B has higher particle density for Lipid II than cavity A over the total simulation time. Therefore, cavity B must be the acceptor site. However, additional high density was obtained near the TM1, which is stronger than the density at cavity A. The authors should elaborate and explain. Additionally, it is not clear what happened to the Und-PP after the reactions. Its fate is not explicitly clarified in the proposed mechanism and the model presented by the authors.

We thank the Reviewer for highlighting this. We have discussed this further as part of our revisions and added 5 sub-figures in Supplementary Fig. 5 to improve our explanation and visualization of the data.

The authors showed that the GT activity of the W102F mutant remains similar. Is the aromatic nature of both amino acids necessary in this case? If so, the W102Y mutant will probably show similar effects. The authors should discuss further the implication of the Trp102 mutations from a mechanistic point of view. Additionally, the authors discuss many mutations in the “RodA active site and mechanism of catalysis” section. It would be insightful to see MD simulations of those mutants and analysis.

We thank the Reviewer for this comment. We also anticipate that the aromatic nature of the residue is required in this position in relation to glycan strand formation, and reasoned a W to F mutation was more sensible to make than a W to Y since that could add a potential, perhaps ‘confusing’ hydrogen bonding interaction. We discuss this residue in the discussion between line 356-362 in relation to an analogous situation with bacterial cellulose synthase from a mechanistic standpoint. We speculate that the role may be in a form of Pi-Pi sugar-amino acid stacking interaction as detailed in the literature (Hudson, *et al*, *Journal of American Chemical Society*, 2015).

The authors presented activity data for many GT mutants in Figure 2e and SI fig 7a. However, mechanistic insight is lacking for these. The authors may try to perform MD simulations of these mutants to gain atomistic insight into the mechanism since that is the manuscript’s focus.

We have interpreted the impact of the mutations, with most of them being a removal of a sidechain, suggesting a mechanistic or binding role. We agree that further QM/MM calculations to assess the impact of these residues on mechanism represents an exciting future direction for the continuation of this project, and thank the Reviewer for the suggestion.

The length of the simulations may be inadequate to capture the conformational dynamics relevant to the overall reaction. As pointed out by the authors, the MD simulations did not capture the 10-degree vertical tilt of PBP2 with respect to the bilayer. The MD simulations only show that they are “dynamic .” I would suggest longer MD simulations and enhanced MD simulations for better conformational sampling.

Enhanced sampling and longer simulations are beyond the scope of this study. We already include 10s of μ s of simulation data within this manuscript that illustrate the potential conformational rearrangement of the PBP2 domain.

The authors stated, “We suggest that Arg210 in cavity A and Arg48 in cavity B, as well as the arginine residues in PL2 (97-111) (Arg101 and Arg109), could facilitate this mechanism through coordination of the phosphate head groups.” ♦ *The authors should provide more quantifiable metrics and details.*

We have used PyLipID to further quantify occupancy times for both cavities and the residues within then for the CGMD simulations.

I am not clear on how the DFTB calculations provide any insight. In the calculations, the authors connected two states by the Nudged Elastic Band method. However, this does not confirm the feasibility of the reaction. The authors should discuss the energy along the optimized NEB path to ensure feasibility. The movie does not provide enough information either.

The purpose of the DFTB calculations was to show that the overall geometry was mechanistically sensible, rather than to extract energies. We have clarified this in the text.

Minor:

1. Definite of PL2 is missing in SI Fig 5b

We have added an orange box in Supplementary Fig. 5b to highlight PL2.

2. Reference 5b and 5c is switched in the main text in line 310 and 319

We have not changed these, as we believe we are referencing the correct figures.

3. It is very difficult to distinguish two conformations from Figure 5a.

We have changed the color of one of the conformations to orange to make it easier to distinguish between the two conformations.

4. Figure 6 could be annotated better with relevant information. For example, what is the groove the authors referred to in the main text?

We have expanded this in the figure legend, as we thought that adding more text to the image would cause some unwarranted confusion.

5. The authors stated, “the tight coordination of the polyprenyl tails in both cavities A and B anchors the nascent PG to the membrane (Supplementary Fig. 10a).” This is unclear in Figure 10a. It only shows the growth of the PG and RMSF. The “tight coordination” is missing. Similarly, Figure 10b does not show the secondary structure analysis, although the text claimed, “secondary structure of RodA-PBP2 is stable during the simulations (Supplementary Fig. 10b).”

We have edited this text and updated the references to the correct figures. We thank the Reviewer for highlighting this.

REVIEWERS' COMMENTS

Reviewer #2 (Remarks to the Author):

The authors have addressed all of my points and revised the manuscript accordingly, and I would like to congratulate them to this important piece of work. There is one remaining issue about the question whether or not the TM of PBP2 is needed for activity of RodA. In my view it is important to clarify this point because the information given in the rebuttal is not consistent with the data shown in Fig. S1.

1. My previous Major Points 1:

a) I think the panels are swapped in the revised Figure S1 g/h: Unlike what is written in the Figure legend, Fig. 2h shows the protein gel and 2g the activity assay.

b) More important, in their rebuttal they wrote "Indeed, RodA in isolation has very low activity, but a version of the fusion which terminates just after the transmembrane helix of PBP2 has significantly enhanced Lipid II polymerisation activity, which is comparable to the full length RodA-PBP2 product. Therefore, in our opinion we can reliably conclude that the presence of the single transmembrane helix of PBP2 in complex with RodA stimulates lipid II GT activity. ". This cannot be correct because Fig. 2g (lane on the right side) clearly shows that RodA-TM(PBP2) has very little activity, comparable to RodA alone.

In the rebuttal to my next point they repeat their conclusion: " ... new data to the revised manuscript showing that the addition of the PBP2 TM helix enhances Lipid II polymerisation in vitro compared to RodA alone ". However, I don't see this effect when comparing the lanes 'RodA in DDM' with 'RodA-TM(PBP2) in DDM' in S1g. Please clarify.

Please correct the panel/legend label and clarify whether RodA-TM(PBP2) has activity comparable to full-length RodA-PBP2 or RodA alone.

2. My previous Major Points 3: They wrote " Whilst in theory it is possible to quantify the amount of fluorescent Lipid II consumed, in practice this is highly inaccurate, not least because the reaction does not proceed to completion under the conditions used. ". It is not clear to me why they could not quantify the remaining lipid II after the reaction by densitometry of the lipid II band (at the bottom of the gels)? Comparison of the lipid II band intensities to controls (without protein or with inactive protein) should tell how much lipid II has been consumed in each reaction. But I understand that it is semi-quantitative and the information might not be needed for the conclusions.

Reviewer #3 (Remarks to the Author):

The authors have adequately addressed all of my concerns.

Reviewer #4 (Remarks to the Author):

The authors have addressed some of my queries but I am still not convinced that the structure provides new insights on the mechanism. As I mentioned before, a complex with lipid II would bring the new understanding on the lipid II polymerization. My concerns over the use of detergents are still standing and any conclusions should take that into account.

Reviewer #5 (Remarks to the Author):

The authors have satisfactorily answered the questions and concerns. I particularly like the fact that the authors explicitly clarified the limitations of their approach in this version of the manuscript. I recommend publishing the version without any revision.

Response to reviewers

Nygaard et al. “Structural basis of peptidoglycan synthesis by *E. coli* RodA-PBP2 complex”

We appreciate the positive response from all the Reviewers. Reviewers #3 and #5 are satisfied with our revised manuscript and have no additional suggestions. Here is our point-to-point response to the comments from Reviewer #2 and #4.

Reviewer #2:

The authors have addressed all of my points and revised the manuscript accordingly, and I would like to congratulate them to this important piece of work. There is one remaining issue about the question whether or not the TM of PBP2 is needed for activity of RodA. In my view it is important to clarify this point because the information given in the rebuttal is not consistent with the data shown in Fig. S1.

1. My previous Major Points 1:

a) I think the panels are swapped in the revised Figure S1 g/h: Unlike what is written in the Figure legend, Fig. 2h shows the protein gel and 2g the activity assay.

Our apologies, the figure legend is indeed swapped compared to the figures presented and this has been corrected.

b) More important, in their rebuttal they wrote "Indeed, RodA in isolation has very low activity, but a version of the fusion which terminates just after the transmembrane helix of PBP2 has significantly enhanced Lipid II polymerisation activity, which is comparable to the full length RodA-PBP2 product. Therefore, in our opinion we can reliably conclude that the presence of the single transmembrane helix of PBP2 in complex with RodA stimulates lipid II GT activity." This cannot be correct because Fig. 2g (lane on the right side) clearly shows that RodA-TM(PBP2) has very little activity, comparable to RodA alone.

Close inspection of the results of this experiment shows that the RodA-TM(PBP2) construct does have significantly more GT activity than RodA alone, albeit at a lower level than the RodA-PBP2 fusion. We now provide an increase sized version of this panel for inspection.

The text used in the paper itself reads: “RodA in isolation has residual GT activity but is significantly stimulated by the presence of the transmembrane helix of PBP2 in both the full-length fusion and the truncated version. These results are consistent with what was previously shown for the *T. thermophilus* proteins (reference 15)”. We use the wording “stimulated by the presence of the transmembrane helix” but do not imply that the level of stimulation is the same in truncated and full-length protein which would require a quantitative measure for which there is no current accurate methodology as discussed previously and below. In our opinion, this is consistent with data in the literature as cited.

In the rebuttal to my next point, they repeat their conclusion: " ... new data to the revised manuscript showing that the addition of the PBP2 TM helix enhances Lipid II polymerisation in vitro compared to RodA alone ". However, I don't see this effect when comparing the lanes 'RodA in DDM' with 'RodA-TM(PBP2) in DDM' in S1g. Please clarify.

Please see above.

Please correct the panel/legend label and clarify whether RodA-TM(PBP2) has activity comparable to full-length RodA-PBP2 or RodA alone.

The Supplementary Figure legend for 1g and 1h have been swapped as directed and again apologies for this mistake.

As we state above, in our opinion the wording is now clear in the text of the paper and consistent with Supplementary Figure 1g

2. My previous Major Points 3: They wrote " Whilst in theory it is possible to quantify the amount of fluorescent Lipid II consumed, in practice this is highly inaccurate, not least because the reaction does not proceed to completion under the conditions used. ". It is not clear to me why they could not quantify the remaining lipid II after the reaction by densitometry of the lipid II band (at the bottom of the gels)? Comparison of the lipid II band intensities to controls (without protein or with inactive protein) should tell how much lipid II has been consumed in each reaction. But I understand that it is semi-quantitative and the information might not be needed for the conclusions.

We appreciate the Reviewer's interest in this aspect of our assay but are inclined to maintain our position that whilst the gel-based assay is informative on overall GT activity and the length of Lipid II polymers produced, this approach is largely and reliably qualitative but not truly quantitative. Densitometry of the remaining Lipid II is indeed possible but given these results presented are from single time point incubation properly meaningful quantitative information is, in our opinion, questionable. We note that in a recent paper by Shlosman et al (Nat Commun 2023 Jun 10;14(1):3439. doi: 10.1038/s41467-023-39037-9) "quantification" of Alexa Fluor 488 labelled Lipid II polymerisation was attempted using densitometry as presented in Figure 4, panel A and D. The quantification is presented as "normalised product" and "calculated by quantifying the total intensity of glycan chains, subtracting the background (Lipid II-only control) and normalizing by the average intensity of the corresponding WT control". In the corresponding gels in this paper, remaining Lipid II is only partially presented so it is difficult to independently assess the validity of the approach. We do not wish to be overly critical of the work presented in this paper, but instead would like to use this as an example of how difficult it is to provide a quantitative measure by a densitometry approach.

Finally, we are currently working on an alternative fluorescence based quantitative assay which we anticipate will be deployed and utilized in subsequent studies on these and other Lipid II polymerisation enzymes.

Reviewer #4:

The authors have addressed some of my queries but I am still not convinced that the structure provides new insights on the mechanism. As I mentioned before, a complex with lipid II would bring the new understanding on the lipid II polymerization. My concerns over the use of detergents are still standing and any conclusions should take that into account.

We thank Reviewer #4 for acknowledging that we addressed some of their concerns. We are unfortunately not able to present a structure of the complex with Lipid II bound. This is an experiment which we have attempted but alas have thus far been unsuccessful. We expect to continue this work and hope to be able to present a structure of the complex in a future publication.